# Optimizing Generalized Rate Metrics with Three Players

**Harikrishna Narasimhan, Andrew Cotter, Maya Gupta**
Google Research
1600 Amphitheatre Pkwy, Mountain View, CA 94043
{hnarasimhan, acotter, mayagupta}@google.com

## Abstract

We present a general framework for solving a large class of learning problems with non-linear functions of classification rates. This includes problems where one wishes to optimize a non-decomposable performance metric such as the F-measure or G-mean, and constrained training problems where the classifier needs to satisfy non-linear rate constraints such as predictive parity fairness, distribution divergences or churn ratios. We extend previous two-player game approaches for constrained optimization to an approach with three players to decouple the classifier rates from the non-linear objective, and seek to find an equilibrium of the game. Our approach generalizes many existing algorithms, and makes possible new algorithms with more flexibility and tighter handling of non-linear rate constraints. We provide convergence guarantees for convex functions of rates, and show how our methodology can be extended to handle sums-of-ratios of rates. Experiments on different fairness tasks confirm the efficacy of our approach.

## 1 Introduction

In many real-world machine learning problems, the performance measures used to evaluate a classification model are non-linear functions of the classifier's prediction rates. Examples include the F-measure and G-mean used in class-imbalanced classification tasks [1–5], metrics such as predictive parity used to impose *fairness* goals [6], the win-loss-ratio used to measure classifier *churn* [7], KL-divergence based metrics used in *quantification* tasks [8, 9] and score-based metrics such as the PR-AUC [10]. Because these goals are non-linear and are non-continuous in the model parameters, it becomes very challenging to optimize with them, especially when they are used in constraints [11].

Prior work on optimizing generalized rate metrics has largely focused on unconstrained learning problems. These approaches fall under two broad categories: surrogate-based methods that replace the classifier rates with convex relaxations [12–17], and oracle-based methods that formulate multiple cost-sensitive learning tasks, and solve them using an oracle [18–22, 11]. Both these approaches have notable deficiencies. The first category of methods rely crucially on the surrogates being close approximations to the rate metric, and perform poorly when this is not the case (see e.g. experiments in [20]). The use of surrogates becomes particularly problematic with constrained training problems, as relaxing the constraints with convex upper bounds can result in solutions that are over-constrained or infeasible [23]. The second category of methods assume access to a near-optimal cost-sensitive oracle, which is usually unrealistic in practice.

In this paper, we present a three-player approach for learning problems where both the objective and constraints can be defined by general functions of rates. The three players optimize over model parameters, Lagrange multipliers and slack variables to produce a game equilibrium. Our approach generalizes many existing algorithms (see Table 2), and makes possible new algorithms with more flexibility and tighter handling of non-linear rate constraints. Specifically, we give a new method

(Algorithm 2) that can handle a wider range of performance metrics than previous surrogate methods (such as e.g. KL-divergence based metrics that only take inputs from a restricted range), and can be applied to constrained training problems without the risk of over-constraining the model because it needs to use surrogates less. To our knowledge, this is the first practical, surrogate-based approach that can handle constraints on generalized rate metrics.

We show convergence of our algorithms for objectives and constraints that are *convex* functions of rates. This result builds on previous work by Cotter et al. [23, 24] for handling linear rate constraints, and additionally resolves an unanswered question in their work on the convergence of Lagrangian optimizers for non-zero-sum games. We also extend our framework to develop a heuristic (Algorithm 3) for optimizing performance measures that are a *sum-of-ratios* of rates (e.g. constraints on predictive parity and F-measure), and demonstrate their utility on real-world tasks.

**Related work:** Many fairness goals can be expressed as *linear* constraints on a model's prediction rates [16, 25]. Recent work has focused on optimizing with linear rate constraints by computing an equilibrium of a game between players who optimize model parameters $\theta$ and Lagrange multipliers $\lambda$ [26–28, 23, 24]. Of these, the closest to us is the work of Cotter et al. (2019) [23], who propose the idea of having only the $\theta$-player optimize a surrogate objective, and having the $\lambda$-player use the original rates. We adapt and build on this idea to handle general functions of rates. Other game-based formulations include the approach of Wang et al. [29] for optimizing multivariate evaluation metrics. Their setup is very different from ours, with their players optimizing over (distributions on) all possible labelings of the data. Moreover, they do not handle constraints on the classifier.

There has also been a concentrated effort on optimizing performance measures such as the F-measure that are fractional-linear functions of rates [18–20, 30, 31]. Many of these works exploit the pseudo-convex structure of the F-measure, but this property is absent for the problems that we consider where we need to handle sums or differences of ratios. Pan et al. [32] provide a heuristic approach for unconstrained sums-of-ratios, and recently Celis et al. [33] handle constraints that are sums-of-ratios, but do so by solving a large number of linearly constrained sub-problems, with the number of sub-problems growing exponentially with the number of rates. In contrast, we provide a practical algorithm that handles both objectives and constraints that are sums-of-ratios of rates.

## 2 Problem Setup

Let $\mathcal{X} \subseteq \mathbb{R}^d$ and $\mathcal{Y} = \{\pm 1\}$ be the instance and label spaces. Let $f_\theta : \mathcal{X} \to \mathbb{R}$ be a prediction model, parameterized by $\theta \in \Theta$. Given a distribution $D$ over $\mathcal{X} \times \mathcal{Y}$, we define the model's rate on $D$ as:

$$R(\theta; D) = \mathbf{E}_{(X,Y) \sim D} \left[ \mathbb{I}\{Y = \text{sign}(f_\theta(X))\} \right].$$

For example, if $D_A$ denotes the distribution over a sub-population $A \subseteq X$, then $R(\theta; D_A)$ gives us the accuracy of $f_\theta$ on this sub-population. If $D_+$ denotes a conditional distribution over the positively-labeled examples, then $R(\theta; D_+)$ is the true-positive rate $\text{TPR}(\theta)$ of $f_\theta$ and $1 - R(\theta; D_+)$ is the false-negative rate $\text{FNR}(\theta)$ of $f_\theta$. Similarly, if $D_-$ denotes a conditional distribution over the negatively-labeled examples, then $\text{TNR}(\theta) = R(\theta; D_-)$ is the true-negative rate of $f_\theta$, and $\text{FPR}(\theta) = 1 - R(\theta; D_-)$ is the false-positive rate of $f_\theta$. Further, the true-positive and false-positive proportions are given by $\text{TP}(\theta) = p \, \text{TPR}(\theta)$ and $\text{FP}(\theta) = (1 - p) \, \text{FPR}(\theta)$, where $p = \mathbf{P}(Y = 1)$; we can similarly define the true negative proportion TN and false negative proportion FN.

We will consider several distributions $D_1, \ldots, D_K$ over $\mathcal{X} \times \mathcal{Y}$, and use the short-hand $R_k(\theta)$ to denote the rate function $R(\theta; D_k) \in [0, 1]$. We will denote a vector of $K$ rates as: $\mathbf{R}(\theta) = [R_1(\theta), \ldots, R_K(\theta)]^\top$. We assume we have access to unbiased estimates of the rates $\hat{R}_k(\theta) = \frac{1}{n_k} \sum_{i=1}^{n_k} \mathbb{I}\{y_i^k = \text{sign}(f_\theta(x_i^k))\}$, computed from samples $S = \{(x_1^k, y_1^k), \ldots, (x_{n_k}^k, y_{n_k}^k)\} \sim D_k$.

We will also consider stochastic models which are defined by distributions over the model parameters $\Theta$. Let $\Delta_\Theta$ denote the set of all such distributions over $\Theta$, where for any $\mu \in \Delta_\Theta$, $\mu(\theta)$ is the probability mass on model $\theta$. We define the rate $R_k$ for a stochastic model $\mu$ to be the expected value for random draw of a model from $\mu$, so $R_k(\mu) = \mathbf{E}_{\theta \sim \mu}[R_k(\theta)]$.

**Generalized Rate Metrics as Objective.** In many real-world applications the performance of a model is evaluated by a function of multiple rates: $\psi(R_1(\theta), \ldots, R_K(\theta))$ for some $\psi : [0, 1]^K \to \mathbb{R}$. For example, a common evaluation metric used in class-imbalanced classification tasks is the G-mean: $\text{GM}(\theta) = 1 - \sqrt{\text{TPR}(\theta) \, \text{TNR}(\theta)}$, which is a *convex* function of rates [34, 35]. Similarly, a popular

Table 1: Examples of generalized rate metrics. $\psi$ is either convex (C), pseudo-convex (PC) or a sum or difference of ratios (SR). $p = \mathbf{P}(Y = 1)$ and $\hat{p}$ is the predicted proportion of positives. $wins$ ($losses$) is the fraction of correct (wrong) predictions by the new model among examples where it disagrees with the old model. $A$ and $B$ refer to different protected groups or slices of the population.

| Measure | Definition | $\psi$ | Type |
|---|---|---|---|
| G-mean [34, 35] | $1 - \sqrt{\mathrm{TPR} \times \mathrm{TNR}}$ | $1 - \sqrt{z_1 z_2}$ | C |
| H-mean [36] | $1 - \frac{2}{1/\mathrm{TPR} + 1/\mathrm{TNR}}$ | $1 - \frac{2}{1/z_1 + 1/z_2}$ | C |
| Q-mean [5] | $1 - \sqrt{\mathrm{FPR}^2 + \mathrm{FNR}^2}$ | $1 - \sqrt{z_1^2 + z_2^2}$ | C |
| KLD [8, 9] | $p \log(\frac{p}{\hat{p}}) + (1-p)\log(\frac{1-p}{1-\hat{p}})$ | $p \log(\frac{p}{z_1}) + (1-p)\log(\frac{1-p}{z_2})$ | C |
| F-measure [37] | $1 - \frac{2\mathrm{TP}}{2\mathrm{TP} + \mathrm{FP} + \mathrm{FN}}$ | $1 - \frac{2z_1}{2z_1 + z_2 + z_3}$ | PC |
| Predictive parity [6] | $\frac{\mathrm{TP}_A}{\mathrm{TP}_A + \mathrm{FN}_A} - \frac{\mathrm{TP}_B}{\mathrm{TP}_B + \mathrm{FN}_B}$ | $\frac{z_1}{z_1 + z_2} - \frac{z_3}{z_3 + z_4}$ | SR |
| F-measure parity | $\frac{2\mathrm{TP}_A}{2\mathrm{TP}_A + \mathrm{FP}_A + \mathrm{FN}_A} - \frac{2\mathrm{TP}_B}{2\mathrm{TP}_B + \mathrm{FP}_B + \mathrm{FN}_B}$ | $\frac{2z_1}{2z_1 + z_2 + z_3} - \frac{2z_4}{2z_4 + z_5 + z_6}$ | SR |
| Churn [7] | $\frac{wins_A}{losses_A} - \frac{wins_B}{losses_B}$ | $\frac{z_1}{z_2} - \frac{z_3}{z_4}$ | SR |
| PR-AUC [10] | $1 - \frac{1}{M}\sum_{m=1}^{M} \frac{\mathrm{TP}_m}{\mathrm{TP}_m + \mathrm{FP}_m}$, where $\mathrm{TP}_m, \mathrm{FP}_m$ are evaluated at the largest threshold $\tau$ at which recall of $f_\theta(X) + \tau$ is $\geq \frac{m+0.5}{M}$ | | |

evaluation metric used in text retrieval is the F-measure: $F_1(\theta) = \frac{2\mathrm{TP}(\theta)}{2\mathrm{TP}(\theta) + \mathrm{FP}(\theta) + \mathrm{FN}(\theta)}$, which is a fractional-linear or *pseudo-convex* function of rates [38]. See Table 1 for more examples. One can consider directly optimizing these performance measures during training:

$$\min_{\theta \in \Theta} \psi\left(R_1(\theta), \ldots, R_K(\theta)\right). \tag{P1}$$

**Generalized Rate Metrics as Constraints.** There are also many applications, where one wishes to impose constraints defined by non-linear function of rates, for example to ensure group-specific fairness metrics. Examples include: (i) *Predictive parity fairness*: Fair classification tasks where one wishes to match the *precision* of a model across different protected groups [6]: $\left| \frac{\mathrm{TP}_A(\theta)}{\mathrm{TP}_A(\theta) + \mathrm{FP}_A(\theta)} - \frac{\mathrm{TP}_B(\theta)}{\mathrm{TP}_B(\theta) + \mathrm{FP}_B(\theta)} \right| \leq \epsilon$. One may also want to match e.g. the *F-measure* of a model across different groups. (ii) *Distribution matching*: Fairness or quantification tasks where one wishes to match the distribution of a model's outputs across different protected groups [9, 15]. One way to achieve this is to constrain the *KL-divergence* between the overall class proportion $p$ and proportion of predicted positives for each group $\hat{p}_A$ [11], i.e. to enforce $\mathrm{KLD}(p, \hat{p}_A) \leq \epsilon$, where KLD is convex in $\hat{p}_A$. (iii) *Churn*: Problems where one wishes to replace a legacy model with a more accurate model while limiting the changed predictions between the new and old models (possibly across different slices of the user base). This is ideally framed as constraints on the win-loss ratio's [24], which can be expressed as ratios of rates [7]. These and related problems can be framed generally as:

$$\min_{\theta \in \Theta} g(\theta) \quad \text{s.t.} \quad \phi^j\left(R_1(\theta), \ldots, R_K(\theta)\right) \leq 0, \quad \forall j \in [J], \tag{P2}$$

for some objective function $g : \Theta \to \mathbb{R}$, and $J$ constraint functions $\phi^j : [0,1]^K \to \mathbb{R}$. We also consider a special case of (P2) with an objective and a constraint that is a sum-of-ratios of rates:

$$\min_{\theta \in \Theta} \sum_{m=1}^{M} \frac{\langle \boldsymbol{\alpha}_m, \mathbf{R}(\theta) \rangle}{\langle \boldsymbol{\beta}_m, \mathbf{R}(\theta) \rangle} \quad \text{s.t.} \quad \sum_{m=M+1}^{2M} \frac{\langle \boldsymbol{\alpha}_m, \mathbf{R}(\theta) \rangle}{\langle \boldsymbol{\beta}_m, \mathbf{R}(\theta) \rangle} \leq \gamma, \tag{P3}$$

for coefficients $\boldsymbol{\alpha}_m, \boldsymbol{\beta}_m \in \mathbb{R}_+^K$, $\forall m \in [2M]$ and slack $\gamma \in \mathbb{R}$.

Our setup can also be used to optimize score-based metrics, such as the area under the precision-recall curve (PR-AUC), that summarize the performance of a score model $f_\theta : \mathcal{X} \to \mathbb{R}$ across multiple thresholds. We use the approach of Eban et al. [10] to (approximately) express PR-AUC as a Riemann summation of the precision of $f_\theta$ at thresholds $\tau_1, \ldots, \tau_M \in \mathbb{R}$ at which the recall of $f_\theta$ is $\frac{0.5}{M}, \frac{1.5}{M}, \ldots, \frac{M-0.5}{M}$ resp. This results in a formulation similar to (P3). See Appendix E for details.

We next provide a three-player framework for solving (P1)–(P3). We note that our formulations can be equivalently regarded as *two-player* games where one player is in charge of the parameters that need to be minimized, and the other player is in charge of the parameters that need to be maximized. We however find the three-player viewpoint to be a useful way to think about the problem algorithmically in that the three sets of optimization parameters can use different algorithms (see Table 2).

Table 2: Algorithms for (P1)–(P3) with 3 players. Frank-Wolfe [20], SPADE [14], NEMSIS [15] are previous algorithms. Alg. 1–3 are the proposed methods. Each player can do Best Response (BR), Online Gradient Descent (OGD) or Follow-The-Leader (FTL), and the game is zero-sum (ZS) or not. The first five algorithms find an approximate Nash or Coarse-Correlated (C.C.) equilibrium assuming $\psi$ and $\phi^j$'s are convex. Since (P3) is non-convex in the rates, Alg. 3 may not find an equilibrium.

| Alg. | P | Player Objective | | | Player Strategy | | | ZS | Equil |
|------|---|---|---|---|---|---|---|---|---|
| | | $\xi$ | $\lambda$ | $\theta$ | $\xi$ | $\lambda$ | $\theta$ | | |
| F-W | P1 | - | $\left(\min_{\xi}\mathcal{L}_1(\xi;\cdot)\right) + \mathcal{L}_2(\theta;\cdot)$ | $\mathcal{L}_2$ | - | FTL | BR | ✓ | Nash |
| SPADE | P1 | $\mathcal{L}_1$ | $\mathcal{L}_1 + \tilde{\mathcal{L}}_2$ | $\tilde{\mathcal{L}}_2$ | BR | OGD | OGD | ✓ | Nash |
| NEMSIS | P1 | - | $\left(\min_{\xi}\mathcal{L}_1(\xi;\cdot)\right) + \tilde{\mathcal{L}}_2(\theta;\cdot)$ | $\tilde{\mathcal{L}}_2$ | - | FTL | OGD | ✓ | Nash |
| Alg. 1 | P1-2 | $\mathcal{L}_1$ | $\mathcal{L}_1 + \mathcal{L}_2$ | $\mathcal{L}_2$ | BR | OGD | BR | ✓ | Nash |
| Alg. 2 | P1-2 | $\mathcal{L}_1$ | $\mathcal{L}_1 + \mathcal{L}_2$ | $\tilde{\mathcal{L}}_2$ | BR | OGD | OGD | ✗ | C. C. |
| Alg. 3 | P3 | $\mathcal{L}_1$ | $\mathcal{L}_1 + \mathcal{L}_2$ | $\tilde{\mathcal{L}}_2$ | OGD | OGD | OGD | ✗ | - |

## 3 Generalized Rate Metric Objective

We first present algorithms for the unconstrained problem in (P1). We assume $\psi$ is strictly convex, and is $L$-Lispchitz w.r.t. the $\ell_\infty$-norm. For simplicity, we assume that $\psi$ is monotonically increasing in all arguments and that $\nabla\psi(\mathbf{0}) = \mathbf{0}$, although our approach easily extends to more general metrics that are e.g. monotonically increasing in some arguments and monotonically decreasing in others.

**Game Formulation.** We equivalently re-write (P1) to de-couple the rates $R_k$ from the non-linear function $\psi$ by introducing auxiliary variables $\xi_1, \ldots, \xi_K \in [0, 1]$:

$$\min_{\theta\in\Theta,\,\xi\in[0,1]^K} \psi\left(\xi_1,\ldots,\xi_K\right) \text{ s.t. } R_k(\theta) \leq \xi_k, \ \forall k \in [K]. \tag{1}$$

A standard approach for solving (1) is to write the Lagrangian for the problem with Lagrange multipliers $\lambda_1, \ldots, \lambda_K \in \mathbb{R}_+$ for the $K$ constraints:

$$\mathcal{L}(\theta, \xi; \lambda) = \psi(\xi) + \sum_{k=1}^{K}\lambda_k\left(R_k(\theta) - \xi_k\right) = \underbrace{\psi(\xi) - \sum_{k=1}^{K}\lambda_k\xi_k}_{\mathcal{L}_1(\xi;\lambda)} + \underbrace{\sum_{k=1}^{K}\lambda_k\,R_k(\theta)}_{\mathcal{L}_2(\theta;\lambda)}.$$

Then one maximizes the Lagrangian over $\lambda \in \mathbb{R}_+^K$, and minimizes it over $\theta \in \Theta$ and $\xi \in [0, 1]^K$:

$$\max_{\lambda\in\mathbb{R}_+^K}\ \min_{\substack{\theta\in\Theta\\ \xi\in[0,1]^K}}\ \mathcal{L}_1(\xi;\lambda) + \mathcal{L}_2(\theta;\lambda), \tag{2}$$

Notice that $\mathcal{L}_1$ is convex in $\xi$ (by convexity of $\psi$), while $\mathcal{L}_1$ and $\mathcal{L}_2$ are linear in $\lambda$. We pose this max-min problem as a zero-sum game played with three players: a player who minimizes $\mathcal{L}_1 + \mathcal{L}_2$ over $\theta$, a player who minimizes $\mathcal{L}_1 + \mathcal{L}_2$ over $\xi$, and a player who maximizes $\mathcal{L}_1 + \mathcal{L}_2$ over $\lambda$. Each of the three players can now use different optimization algorithms customized for their problem. If additionally the Lagrangian was convex in $\theta$, one could solve for an equilibrium of this game and obtain a solution for the primal problem (1). However, since $\mathcal{L}_2$ is a weighted sum of rates $R_k(\theta)$, it need not be convex (or even continuous) in $\theta$.

To overcome this difficulty, we expand the solution space from deterministic models $\Theta$ to stochastic models $\Delta_\Theta$ [26, 23, 24, 39], and re-formulate (2) as a problem that is linear in $\mu$, by replacing each $R_k(\theta)$ with $E_{\theta\sim\mu}[R_k(\theta)]$ in $\mathcal{L}_2$:

$$\max_{\lambda\in\Lambda}\ \min_{\substack{\mu\in\Delta_\Theta\\ \xi\in[0,1]^K}}\ \mathcal{L}_1(\xi;\lambda) + \mathcal{L}_2(\mu;\lambda). \tag{3}$$

Here for technical reasons, we restrict the Lagrange multipliers to a bounded set $\Lambda = \{\lambda \in \mathbb{R}_+ \mid \|\lambda\|_1 \leq \kappa\}$; we will choose the radius $\kappa > 0$ later in our theoretical analysis. By solving for an equilibrium of this expanded max-min problem, we can find a stochastic model $\mu \in \Delta_\theta$ that minimizes $\psi(R_1(\mu), \ldots, R_K(\mu))$.

There are two approaches that we can take to find an equilibrium of the expanded game. The first approach is to assume access to an oracle that can perform the minimization of $\mathcal{L}_2$ over $\mu$ for a fixed $\lambda$ and $\xi$. Since this is a linear optimization over the simplex, this amounts to performing a minimization over deterministic models in $\Theta$. The second and more realistic approach is to work with surrogates for the rates that are continuous and differentiable in $\theta$. Let $\tilde{R}_1, \ldots, \tilde{R}_K : \Theta \to \mathbb{R}$ be differentiable convex surrogate functions that are upper bounds on the rates: $R_k(\theta) \leq \tilde{R}_k(\theta), \forall \theta \in \Theta$. We assume access to unbiased stochastic sub-gradients for the surrogates $\nabla_\theta \tilde{R}_k(\theta)$ with $\mathbf{E}[\nabla_\theta \tilde{R}_k(\theta)] \in \partial_\theta \tilde{R}_k(\theta)$. We then define a surrogate-based approximation for $\mathcal{L}_2$:

$$\tilde{\mathcal{L}}_2(\theta; \lambda) = \sum_{k=1}^{K} \lambda_k \, \tilde{R}_k(\theta). \tag{4}$$

All we need to do now is to choose the objective that each player seeks to optimize (true or surrogate [23]) and the strategy that they use to optimize their objective, so that the players converge to an equilibrium of the game. Each of these choices lead to a different algorithm for (approximately) solving (P1). Table 2 summarizes different choices of strategies and objectives for the players, and the type of equilibrium and the algorithm that results from these choices. As we shall see shortly (and also elaborate in Appendix B.1), many existing algorithms can be seen as instances of this template.

**Oracle-based Lagrangian Optimizer.** As a warm-up illustration of our approach, we first describe an idealized algorithm assuming access to an oracle that can approximately optimize $\mathcal{L}_2$ over $\Theta$ (this oracle essentially optimizes a weighted-sum of rates, i.e. a cost-sensitive error objective [40]):

**Definition 1.** *A $\rho$-approximate cost-sensitive optimization (CSO) oracle takes $\lambda$ as input and outputs a model $\theta^* \in \Theta$ such that $\mathcal{L}_2(\theta^*; \lambda) \leq \min_{\theta \in \Theta} \mathcal{L}_2(\theta; \lambda) + \rho$.*

We have all three players optimize the true Lagrangians, with the $\theta$-player and $\xi$-player playing best responses to the opponents strategies, i.e. they perform full optimization over their parameter space, and the $\lambda$-player running online gradient descent (OGD), an algorithm with no-regret guarantees [41]. The $\theta$-player performs best response by using the above oracle to approximately minimize $\mathcal{L}_2$ over $\theta$. For the $\xi$-player, the best response optimization can be computed in closed-form:

**Lemma 1.** *Let $\psi^* : \mathbb{R}_+^K \to \mathbb{R}$ denote the Fenchel conjugate of $\psi$. Then for any $\lambda$ s.t. $\|\lambda\|_1 \leq L$:*

$$\nabla \psi^*(\lambda) \in \mathrm{argmin}_{\xi \in [0,1]^K} \, \mathcal{L}_1(\xi; \lambda).$$

The resulting algorithm outlined in Algorithm 1 is guaranteed to find an approximate Nash equilibrium of the max-min game in (3), and yields an approximate solution to (P1):

**Theorem 1.** *Let $\theta^1, \ldots, \theta^T$ be the iterates generated by Algorithm 1 for (P1), and let $\bar{\mu}$ be a stochastic model with a probability mass of $\frac{1}{T}$ on $\theta^t$. Define $B_\lambda \geq \max_t \|\nabla_\lambda \mathcal{L}(\xi^t, \theta^t; \lambda^t)\|_2$. Then setting $\kappa = L$ and $\eta = \frac{L}{B_\lambda \sqrt{2T}}$, we have w.p. $\geq 1 - \delta$ over draws of stochastic gradients:*

$$\psi\left(\mathbf{R}(\bar{\mu})\right) \leq \min_{\mu \in \Delta_\Theta} \psi\left(\mathbf{R}(\mu)\right) + \mathcal{O}\left(\sqrt{\frac{\log(1/\delta)}{T}}\right) + \rho.$$

**Remark 1** (**Frank-Wolfe: a special case**). *A previous oracle-based approach for optimizing convex functions of rates is the Frank-Wolfe based method of Narasimhan et al. [14]. This method can be recovered from our framework by reformulating the $\lambda$-player's objective to include the minimization over $\xi$: $\mathcal{L}_{FW}(\lambda, \theta) = \min_\xi \mathcal{L}_1(\xi; \lambda) + \mathcal{L}_2(\theta; \lambda)$ and having the $\lambda$-player play the Follow-The-Leader (FTL) algorithm [42] to maximize this objective over $\lambda$. As before, the $\theta$-player plays best response on $\mathcal{L}_2$ using the CSO oracle. See Appendix B.1 for details, where we use recent connections between the classical Frank-Wolfe technique and equilibrium computation [43].*

**Surrogate-based Lagrangian Optimizer.** While the CSO oracle may be available in some special cases (e.g. when $\Theta$ is finite or when the underlying conditional-class probabilities can be estimated accurately), in many practical scenarios, it is not realistic to assume access to an oracle that can optimize non-continuous rates. We now provide a more practical algorithm for solving (P1) where the $\theta$-player optimizes the surrogate Lagrangian function $\tilde{\mathcal{L}}_2$ in (4) instead of $\mathcal{L}_2$ using stochastic gradients $\nabla_\theta \tilde{\mathcal{L}}_2(\theta; \lambda)$. The $\xi$- and $\lambda$-players, however, continue to operate on the true Lagrangian functions $\mathcal{L}_1$ and $\mathcal{L}_2$, which are continuous in the parameters $\xi$ and $\lambda$ that these players optimize.

In our proposed approach, outlined in Algorithm 2, both the $\theta$-player and $\lambda$-player now run online gradient descent algorithms, while the $\xi$-player plays its best response at each iteration. Since it is

| **Algorithm 1** Oracle-based Optimizer | **Algorithm 2** Surrogate-based Optimizer |
|---|---|
| Initialize: $\lambda^0$ | Initialize: $\theta^0, \lambda^0$ |
| **for** $t = 0$ to $T - 1$ **do** | **for** $t = 0$ to $T - 1$ **do** |
|    **if** (P1) **then** |    **if** (P1) **then** |
|      $\xi^t = \nabla\psi^*(\lambda^t)$ |      $\xi^t = \nabla\psi^*(\lambda^t)$ |
|    **else if** (P2) **then** |    **else if** (P2) **then** |
|      $\xi^t \in \operatorname{argmin}_{\xi \in [0,1]^K} \mathcal{L}_1(\xi; \lambda^t)$ |      $\xi^t \in \operatorname{argmin}_{\xi \in [0,1]^K} \mathcal{L}_1(\xi; \lambda^t)$ |
|    **end if** |    **end if** |
|    $\theta^t \in \operatorname{argmin}_{\theta \in \Theta} \mathcal{L}_2(\theta; \lambda^t)$    [CSO] |    $\theta^{t+1} \leftarrow \Pi_\Theta(\theta^t - \eta_\theta \nabla_\theta \tilde{\mathcal{L}}_2(\theta^t; \lambda^t))$ |
|    $\lambda^{t+1} = \Pi_\Lambda(\lambda^t + \eta \nabla_\lambda \mathcal{L}(\theta^t, \xi^t; \lambda^t))$ |    $\lambda^{t+1} \leftarrow \Pi_\Lambda(\lambda^t + \eta_\lambda \nabla_\lambda \mathcal{L}(\theta^t, \xi^t; \lambda^t))$ |
| **end for** | **end for** |
| **return** $\theta^1, \ldots, \theta^T$ | **return** $\theta^1, \ldots, \theta^T$ |

Figure 1: Optimizers for the unconstrained problem (P1) and constrained problem (P2). Here $\Pi_\Lambda$ denotes the $\ell_1$-projection onto $\Lambda$ and $\Pi_\Theta$ denotes the $\ell_2$-projection onto $\Theta$. We denote a (stochastic) gradient of $\mathcal{L}$ by $\nabla_\lambda \mathcal{L}(\theta^t, \xi^t; \lambda^t) = \sum_{k=1}^K (\hat{R}_k(\theta^t) - \xi_k^t)$, where $\hat{R}_k(\theta^t)$ is an unbiased estimate of $R_k(\theta^t)$. We denote a (stochastic) sub-gradient of $\tilde{\mathcal{L}}_2$ by $\nabla_\theta \tilde{\mathcal{L}}_2(\theta^t; \lambda^t) = \sum_{k=1}^K \lambda_k \nabla_\theta \tilde{R}_k(\theta^t)$.

the $\theta$-player alone who optimizes a surrogate, the resulting game between the three players is no longer zero-sum. Yet, we are able to show that the player strategies converge to an approximate *coarse-correlated* (C. C.) equilibrium of the game, and yields an approximate solution to (P1).

**Theorem 2.** *Let $\theta^1, \ldots, \theta^T$ be the iterates of Algorithm 2 for (P1), and let $\bar{\mu}$ be a stochastic model with probability $\frac{1}{T}$ on $\theta^t$. Let $\tilde{\Theta}$ be a convex set and $\tilde{\Theta} = \{\theta \in \Theta \,|\, \tilde{\mathbf{R}}(\theta) \in [0,1]^K\}$. Let $B_\Theta \geq \max_{\theta \in \Theta} \|\theta\|_2$, $B_\theta \geq \max_t \|\nabla_\theta \tilde{\mathcal{L}}_2(\theta^t; \lambda^t)\|_2$ and $B_\lambda \geq \max_t \|\nabla_\lambda \mathcal{L}(\xi^t, \theta^t; \lambda^t)\|_2$. Setting $\kappa = L$, $\eta_\theta = \frac{B_\Theta}{B_\theta \sqrt{T}}$ and $\eta_\lambda = \frac{L}{B_\lambda \sqrt{2T}}$, we have w.p. $\geq 1 - \delta$ over draws of stochastic gradients:*

$$\psi\big(\mathbf{R}(\bar{\mu})\big) \leq \min_{\theta \in \tilde{\Theta}} \psi\big(\tilde{\mathbf{R}}(\theta)\big) + \mathcal{O}\bigg(\sqrt{\frac{\log(1/\delta)}{T}}\bigg).$$

Note the right-hand side contains the optimal value for the surrogate objective $\psi(\tilde{\mathbf{R}}(\cdot))$ and not for the original performance metric. This is unsurprising given the $\theta$-player's inability to work with the true rates. Also, while Algorithm 2 can be applied to optimize over a general (bounded) convex model class $\Theta$, the comparator for our guarantee is a subset of models $\tilde{\Theta} \subseteq \Theta$ for which $\psi(\tilde{\mathbf{R}}(\theta))$ is defined. This is needed as the surrogate $\tilde{\mathbf{R}}(\theta)$ may output values outside the domain of $\psi$.

**Remark 2** (**SPADE, NEMSIS: special cases of our approach**). *Our approach includes two previous surrogate-based algorithms as special cases: SPADE [20] and NEMSIS [15]. SPADE can be recovered from our framework by having the same player strategies as Algorithm 2 but with both the $\theta$- and $\lambda$-players optimizing surrogate objectives, i.e. with the $\theta$-player minimizing $\tilde{\mathcal{L}}_2$ and the $\lambda$-player maximizing $\mathcal{L}_1 + \tilde{\mathcal{L}}_2$. NEMSIS also uses surrogates for both the $\theta$ and $\lambda$ updates. It can be recovered by having the $\theta$-player run OGD on $\tilde{\mathcal{L}}_2$, and having the $\lambda$-player play FTL over $\lambda$ on the combined objective $\min_\xi \mathcal{L}_1(\xi; \lambda) + \tilde{\mathcal{L}}_2(\theta; \lambda)$. See Appendix B.1 for details.*

**Remark 3** (**Application to wider range of metrics**). *Because of their strong reliance on surrogates, SPADE and NEMSIS cannot be applied directly to functions $\psi$ that take inputs from a restricted range (e.g. KL-divergence), unless the surrogates are also bounded in the same range. In Appendix D.1, we point out scenarios where the NEMSIS method [15] fails to optimize the KL-divergence metric, unless the model is sufficiently regularized to not output large negative values. Algorithm 2 has no such restriction and can be applied even if the outputs of the surrogates are not within the domain of $\psi$. This is because it uses the original rates for updates on $\lambda$. As a result, the game play between $\xi$ and $\lambda$ never produces values that are outside the domain of $\psi$.*

## 4   Generalized Rate Metric Constraints

We next describe how to apply our approach to the constrained optimization problem in (P2) and to the special case in (P3). We start with (P2) assuming that the constraints $\phi^j$'s are jointly convex,

monotonic in each argument and $L$-Lipschitz w.r.t. the $\ell_\infty$-norm, and $g$ is a bounded convex function. For convenience, we assume that the $\phi^j$'s are monotonically increasing in all arguments. Constraints on the KL-divergence and G-mean metrics are examples of this setting.

We introduce a set auxiliary variables $\xi_1, \ldots, \xi_K$ for the $K$ rate functions and re-write (P2) as:

$$\min_{\theta \in \Theta, \, \xi \in [0,1]^K} g(\theta)$$
$$\text{s.t.} \quad \phi^j(\xi_1, \ldots, \xi_K) \leq 0, \; \forall j \in [J], \quad R_k(\theta) \leq \xi_k, \; \forall k \in [K].$$

The Lagrangian for the re-written problem is given below, where $\lambda_1, \ldots, \lambda_J \in \mathbb{R}_+$ and $\lambda_{J+1}, \ldots, \lambda_{J+K} \in \mathbb{R}_+$ are the Lagrange multipliers for the two sets of constraints:

$$\mathcal{L}(\theta, \xi; \lambda) = \underbrace{\sum_{j=1}^{J} \lambda_j \, \phi^j(\xi) - \sum_{k=1}^{K} \lambda_{J+k} \, \xi_k}_{\mathcal{L}_1(\xi;\lambda)} + \underbrace{g(\theta) + \sum_{k=1}^{K} \lambda_{J+k} \, R_k(\theta)}_{\mathcal{L}_2(\theta;\lambda)}. \tag{5}$$

As before, we expand the search space to include stochastic models in $\Delta_\Theta$, restrict the Lagrange multipliers to a bounded set $\Lambda = \left\{ \lambda \in \mathbb{R}_+^{J+K} \, \middle| \, \|\lambda\|_1 \leq \kappa \right\}$, and formulate a max-min problem:

$$\max_{\lambda \in \Lambda} \min_{\mu \in \Delta_\Theta, \, \xi \in [0,1]^K} \mathcal{L}_1(\xi; \lambda) + \mathcal{L}_2(\mu; \lambda). \tag{6}$$

One can now apply Algorithm 1 and 2 to this problem. For Algorithm 1, the $\theta$-player uses the CSO oracle to optimize the true Lagrangian $\mathcal{L}_2$, and for Algorithm 2, the $\theta$-player uses OGD to optimize a surrogate Lagrangian $\tilde{\mathcal{L}}_2(\mu; \lambda) = \sum_{k=1}^{K} \lambda_{J+k} \, \tilde{R}_k(\mu)$. In both cases, the $\xi$-player plays best response by minimizing $\mathcal{L}_1$ over $\xi$ using an analytical solution (where available) or using a convex optimization solver.

**Theorem 3.** *Let $\theta^1, \ldots, \theta^T$ be the iterates generated by Algorithm 1 for (P2), and let $\bar{\mu}$ be a stochastic model with a probability mass of $\frac{1}{T}$ on $\theta^t$. Suppose there exists a $\mu' \in \Delta_\Theta$ such that $\phi^j(\mathbf{R}(\mu')) \leq -\gamma, \, \forall j \in [J]$, for some $\gamma > 0$. Let $B_g = \max_{\theta \in \Theta} g(\theta)$. Let $\mu^* \in \Delta_\Theta$ be such that $\mu^*$ is feasible, i.e. $\phi^j(\mathbf{R}(\mu^*)) \leq 0, \, \forall j \in [J]$, and $\mathbf{E}_{\theta \sim \mu^*}[g(\theta)] \leq \mathbf{E}_{\theta \sim \mu}[g(\theta)]$ for every $\mu \in \Delta_\Theta$ that is feasible. Let $B_\lambda \geq \max_t \|\nabla_\lambda \mathcal{L}(\xi^t, \theta^t; \lambda^t)\|_2$. Then setting $\kappa = \frac{2(L+1)B_g}{\gamma}$ and $\eta = \frac{\kappa}{B_\lambda \sqrt{2T}}$, we have w.p. $\geq 1 - \delta$ over draws of stochastic gradients:*

$$\mathbf{E}_{\theta \sim \bar{\mu}}[g(\theta)] \leq \mathbf{E}_{\theta \sim \mu^*}[g(\theta)] + \mathcal{O}\left(\sqrt{\frac{\log(1/\delta)}{T}} + \rho\right) \quad \text{and} \quad \phi^j(\mathbf{R}(\bar{\mu})) \leq \mathcal{O}\left(\sqrt{\frac{\log(1/\delta)}{T}} + \rho\right), \forall j.$$

We have thus shown that Algorithm 1 outputs a stochastic model that has an objective close to the optimal feasible solution for (P2), while also closely satisfying the constraints.

**Theorem 4.** *Let $\theta^1, \ldots, \theta^T$ be the iterates of Algorithm 1 for (P2), and let $\bar{\mu}$ be a stochastic model with prob. $\frac{1}{T}$ on $\theta^t$. Let $\Theta$ be convex and $\tilde{\Theta} = \left\{ \theta \in \Theta \, \middle| \, \forall j, \, \tilde{\mathbf{R}}(\theta) \in [0,1]^K \right\}$. Let $\tilde{\theta}^* \in \tilde{\Theta}$ be such that it satisfies the surrogate-relaxed constraints $\phi^j(\tilde{\mathbf{R}}(\tilde{\theta}^*)) \leq 0, \, \forall j \in [J]$, and $g(\tilde{\theta}^*) \leq g(\theta)$ for every $\theta \in \tilde{\Theta}$ that satisfies the same constraints. Let $B_\Theta \geq \max_{\theta \in \Theta} \|\theta\|_2$, $B_\theta \geq \max_t \|\nabla_\theta \tilde{\mathcal{L}}_2(\theta^t; \lambda^t)\|_2$ and $B_\lambda \geq \max_t \|\nabla_\lambda \mathcal{L}(\xi^t, \theta^t; \lambda^t)\|_2$. Then setting $\kappa = (L+1)T^\omega$ for $\omega \in (0, 0.5)$, $\eta_\theta = \frac{B_\Theta}{B_\theta \sqrt{2T}}$ and $\eta_\lambda = \frac{\kappa}{B_\lambda \sqrt{2T}}$, we have w.p. $\geq 1 - \delta$ over draws of stochastic gradients:*

$$\mathbf{E}_{\theta \sim \bar{\mu}}[g(\theta)] \leq g(\tilde{\theta}^*) + \mathcal{O}\left(\frac{\sqrt{\log(1/\delta)}}{T^{1/2-\omega}}\right) \quad \text{and} \quad \phi^j(\mathbf{R}(\bar{\mu})) \leq \mathcal{O}\left(\frac{\sqrt{\log(1/\delta)}}{T^\omega}\right), \, \forall j.$$

The proof is an adaptation of the analysis in Agarwal et al. (2018) [26] to non-zero-sum games. We point out that despite the $\theta$-player optimizing surrogate functions, the final stochastic model is near-feasible for the original rate metrics. We also note that this result holds even if the surrogates output values outside the domain of the constraint functions $\phi^j$'s (e.g. with KL-divergence constraints). While the above convergence rate is not as good as the standard $\mathcal{O}(1/\sqrt{T})$ rate achievable for OGD, this is similar to the guarantees shown by e.g. Agarwal et al. [26] for linear rate-constrained optimization problems.[1] The reason for the poorer convergence rate is that we are unable to fix the radius $\kappa$ of the space of Lagrange multipliers $\Lambda$ to a constant, and instead set it to a function of $T$.

**Algorithm 3** Surrogate-based Optimizer for (P3)

---

Initialize: $\mathbf{a}^0, \mathbf{b}^0, \theta^0, \lambda^0$
**for** $t = 0$ to $T - 1$ **do**
  $\mathbf{a}^{t+1} = \Pi_{\mathcal{C}}\left(\mathbf{a}^t - \eta_\mathbf{a}\nabla_\mathbf{a}\mathcal{L}_{sr}(\theta^t, \mathbf{a}^t, \mathbf{b}^t; \lambda^t)\right);$
  $\mathbf{b}^{t+1} = \Pi_{\mathcal{C}}\left(\mathbf{b}^t - \eta_\mathbf{b}\nabla_\mathbf{b}\mathcal{L}_{sr}(\theta^t, \mathbf{a}^t, \mathbf{b}^t; \lambda^t)\right),$ where $\mathcal{C} = \{a, b \in \mathbb{R} \mid \underline{a} \le a \le b \le \bar{b}\}$
  $\theta^{t+1} = \Pi_{\Theta}(\theta^t - \eta_\theta \nabla_\theta\tilde{\mathcal{L}}_{sr}(\theta^t; \mathbf{a}^t, \mathbf{b}^t \lambda^t)),$ where $\tilde{\mathcal{L}}_{sr}$ is (7) with $R_k$ replaced with $\tilde{R}_k$
  $\lambda^{t+1} = \Pi_{\Lambda}\left(\lambda^t + \eta_\lambda \nabla_\lambda\mathcal{L}_{sr}(\theta^t, \mathbf{a}^t, \mathbf{b}^t; \lambda^t)\right)$
**end for**
**return** $\theta^1, \dots, \theta^T$

---

**Remark 4** (**Unanswered Question in Cotter et al.**). *Cotter et al. consider optimization problems with linear rate constraints, formulate a non-zero-game like us, and consider two algorithms, one where both the $\theta$- and $\lambda$-player seek to minimize external regret (through OGD updates), and the other where the $\theta$-player optimizes alone minimizes external regret, while the $\lambda$-player minimizes swap regret. They are however able to show convergence guarantees only for a more complicated swap regret algorithm, and leave the analysis of the external regret algorithm unanswered. Theorem 4 provides convergence guarantees for a generalization of their external regret algorithm. In their paper, Cotter et al. do obtain a better $\mathcal{O}(1/\sqrt{T})$ convergence rate for their swap regret algorithm. It is easy to show a similar convergence rate for an adaptation of this algorithm to our setting (see Appendix C), but we stick to our present algorithm because of its simplicity.*

**Case of Sum-of-ratios Metrics.** Moving beyond convex functions of rates, we present a heuristic algorithm for optimizing with objectives and constraints in (P3) that are sums-of-ratios of rates. We assume that the numerators and denominators in each ratio term is bounded, i.e. $\underline{a} \le \langle \boldsymbol{\alpha}_m, \mathbf{R}(\theta) \rangle \le \langle \boldsymbol{\beta}_m, \mathbf{R}(\theta) \rangle \le \bar{b}$, and $\underline{a} \le \langle \boldsymbol{\alpha}'_m, \mathbf{R}(\theta) \rangle \le \langle \boldsymbol{\beta}'_m, \mathbf{R}(\theta) \rangle \le \bar{b}$, $\forall \theta \in \Theta$ for some $\underline{a}, \bar{b} > 0$. Introducing slack variables $a_1, \dots, a_{2M}, b_1, \dots, b_{2M}$ for the numerators and denominators respectively to decouple the rates from the ratio terms, we equivalently re-write (P3):

$$\min_{\substack{\theta \in \Theta \\ \underline{a} \le a_m \le b_m \le \bar{b}}} \sum_{m=1}^{M} \frac{a_m}{b_m} \quad \text{s.t.} \quad \sum_{m=M+1}^{2M} \frac{a_m}{b_m} \le \gamma, \; a_m \ge \langle \boldsymbol{\alpha}_m, \mathbf{R}(\theta) \rangle, \; b_m \le \langle \boldsymbol{\beta}_m, \mathbf{R}(\theta) \rangle, \; \forall m.$$

We then formulate the Lagrangian for the above problem with multipliers $\lambda \in \mathbb{R}_+^{4M+1}$ and get:

$$
\begin{aligned}
\mathcal{L}_{sr}(\theta, \mathbf{a}, \mathbf{b}; \lambda) &= \sum_{m=1}^{M} a_m/b_m + \lambda_0\left(\sum_{m=M+1}^{2M} a_m/b_m - \gamma\right) \\
&\quad + \sum_{m=1}^{2M}\lambda_m(\langle \boldsymbol{\alpha}_m, \mathbf{R}(\theta) \rangle - a_m) + \sum_{m=1}^{2M}\lambda_{2M+m}(b_m - \langle \boldsymbol{\beta}_m, \mathbf{R}(\theta) \rangle).
\end{aligned}
\tag{7}
$$

Because $\mathcal{L}_{sr}$ is non-convex in the slack variables $\mathbf{a}, \mathbf{b}$, strong duality may not hold, and an optimal solution for the dual problem may not be optimal for (P3). Yet, by performing OGD updates for the $\lambda, \theta$ and the slack variables, with the $\theta$-player alone optimizing the surrogate rates $\tilde{R}_k$, we obtain a heuristic to solve (P3). The details are given in Algorithm 3.

## 5 Experiments

We conduct two types of experiments. In the first, we evaluate Algorithm 2 on the task of optimizing a convex rate objective subject to linear rate constraints and show it often performs as well as an oracle-based approach for this problem [11] (and does so without having to make an idealized oracle assumption). In the second, we evaluate Algorithm 3 on the task of optimizing a sum-of-ratios objective subject to sum-of-ratios constraints, and compare it against existing baselines.

**Datasets.** We use five datasets: (1) *COMPAS*, where the goal is to predict recidivism with *gender* as the protected attribute [44]; (2) *Communities & Crime*, where the goal is to predict if a community in the US has a crime rate above the 70th percentile [45], and we consider communities having a black population above the 50th percentile as protected [27]; (3) *Law School*, where the task is to predict whether a law school student will pass the bar exam, with *race* (black or other) as the protected attribute [46]; (4) *Adult*, where the task is to predict if a person's income exceeds 50K/year, with *gender* as the protected attribute [45]; (5) *Wiki Toxicity*, where the goal is to predict if a comment

Table 3: Optimizing KL-divergence fairness metric s.t. error rate constraints. For each method, we report two metrics: A (B), where A is the test fairness metric (*lower* is better) and B is the ratio of the test error rate of the method and that of UncError (*lower* is better). During training, we constrain B to be $\leq 1.1$. Among the last 3 columns, the lowest fairness metric is highlighted in bold.

|  | UncError | PostShift | COCO | Stochastic | Determ. |
|---|---|---|---|---|---|
| COMPAS | 0.115 (1.00) | 0.000 (1.01) | 0.043 (1.01) | **0.000** (1.03) | **0.000** (1.03) |
| Crime | 0.224 (1.00) | 0.005 (1.40) | 0.252 (0.83) | **0.120** (1.11) | 0.146 (1.08) |
| Law | 0.199 (1.00) | 0.001 (1.45) | **0.043** (1.05) | 0.054 (1.12) | 0.056 (1.08) |
| Adult | 0.114 (1.00) | 0.000 (1.22) | **0.011** (1.10) | 0.014 (1.10) | 0.014 (1.10) |
| Wiki | 0.175 (1.00) | 0.001 (1.21) | 0.134 (1.17) | 0.133 (1.09) | **0.127** (1.18) |

Table 4: Optimizing F-measure s.t. F-measure constraints. For each method, we report two metrics: A (B), where A is the overall test F-measure (*higher* is better) and B is the test constraint violation: $\text{Fmeasure}_{prt} - \text{Fmeasure}_{other} - 0.02$ (*lower* is better). The lowest constraint violation is in italic.

|  | UncError | UncF1 | Stochastic | Determ. |
|---|---|---|---|---|
| COMPAS | 0.656 (0.13) | 0.666 (0.09) | 0.627 (*0.07*) | 0.628 (*0.07*) |
| Crime | 0.742 (0.19) | 0.752 (0.19) | 0.711 (*0.11*) | 0.711 (*0.11*) |
| Law | 0.973 (0.10) | 0.975 (0.08) | 0.927 (0.04) | 0.842 (*-0.05*) |
| Adult | 0.675 (*0.03*) | 0.688 (0.06) | 0.660 (0.04) | 0.647 (*0.03*) |
| Wiki | 0.968 (0.18) | 0.967 (0.18) | 0.826 (0.00) | 0.782 (*-0.05*) |

posted on a Wikipedia talk page contains non-toxic/acceptable content, with the comments containing the term '*gay*' considered as a protected group [47]. We use linear models, and hinge losses as surrogates $\tilde{R}_k$. All implementations are in Tensorflow.[2] See Appendix G for additional details.

**KL-divergence Based Fairness Objective.** We consider a demographic-parity style fairness objective that seeks to match the proportion of positives predicted in each group $\hat{p}_G$ with the true proportion of positives $p$ in the data, measured using a KL-divergence metric: $\sum_{G \in \{0,1\}} \text{KLD}(p, \hat{p}_G)$. Note that this is convex in $\hat{p}_G$. We additionally enforce a constraint that the error rate of the model is no more than 10% higher than an unconstrained model that optimizes error rate, i.e. $\hat{err}(f_\theta) \leq 1.1 \, \hat{err}(f_{unc})$. The only previous method that we are aware of that can handle constrained problems of this form is the COCO method of Narasimhan (2018) [11]. This is an oracle-based approach and uses a *plug-in* method to implement the cost-sensitive oracle. We compare Algorithm 2 with COCO, and the post-shift method of Hardt et al. (2016) [48], where we take a pre-trained logistic regression model and assign different thresholds to the groups to correct for fairness disparity [48]. For our algorithm, we report the performance of both the trained stochastic classifier and the best *deterministic* classifier chosen through a 'best iterate' heuristic of Cotter et al. (2019) [23]. The results are shown in Table 3. PostShift performs the best on the fairness metric, but on all datasets except COMPAS, fairs poorly on the constraint. The proposed method and COCO achieve different trade-offs between optimizing the objective and satisfying the error rate constraint. The stochastic classifier trained by our method closely satisfies the constraint on almost all datasets, and yields a lower objective than COCO on three datasets. Moreover, the best deterministic classifier often has a similar performance to the stochastic classifier. We present further analysis and additional comparisons in Appendix G.1.

**F-measure Based Parity Constraints.** We consider the fairness goal of training a classifier that yields at least as high a F-measure for the protected group as it does for the rest of the population, and impose this as a constraint. Specifically, we seek to maximize the overall F-measure subject to the constraint: $\text{Fmeasure}_{prt} \geq \text{Fmeasure}_{other} - 0.02$. We apply Algorithm 3 and compare it with an unconstrained classifier that optimizes error rate (UncError) and a plug-in classifier that optimizes the F-measure without constraints (UncF1). As seen in Table 4, while UncF1 yields a better objective than UncError, both the baselines fail to satisfy the constraint. On four of the datasets, the stochastic classifier trained by our approach yields moderate to significant reduction in constraint violation. On Adult, the trained stochastic classifier yields a very small constraint violation on the training set (0.004), but does not perform as well on the test set. The deterministic classifiers have an equal or lower constraint violation compared to the stochastic classifiers, but at the cost of a lower objective.

## Acknowledgments

We thank Qijia Jiang and Heinrich Jiang for helpful feedback on draft versions of this paper.

## Footnotes

[1] See Theorem 3 in their paper, where, for $T = \mathcal{O}(n^{4\alpha})$ iterations, the error bound is $\tilde{\mathcal{O}}(n^{-\alpha}) = \tilde{\mathcal{O}}(T^{-1/4})$.

[2] https://github.com/google-research/google-research/tree/master/generalized_rates

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
