[Supplementary Material · GeneralizedRateConstraints_NeurIPS2019_CameraReady_supp.pdf]

# Optimizing Generalized Rate Metrics with Three Players
## Appendix

**Outline.** The appendix is structured as follows:

- In Section A, we provide some preliminaries on Fenchel conjugates and online convex optimization.
- In Section B, we highlight connections between our framework and previous approaches to solve (P1) and (P2).
- In Section C, we elaborate on connections between our results and that of Cotter et al. [23].
- In Section D, we highlight the limitations of previous surrogate-based approaches, using the KL-divergence metric as an example.
- In Section E, we described how our framework can be applied to optimize PR-AUC and other precision-recall metrics.
- In Section F, we describe an adaptation of the SPADE algorithm [14] to solve (P2).
- In Section G, we provide additional details about our experiments, and additional comparisons with our extension of the SPADE algorithm.
- In Sections H–K, we provide proofs for all the theorems in the main text.

## A  Preliminaries

We cover some preliminary material that will be useful in subsequent sections.

**Fenchel conjugate**  For a convex function $\psi : \mathbb{R}^K \to \mathbb{R}$, the Fenchel conjugate $\psi^* : \mathbb{R}^K \to \mathbb{R}$ is defined by:

$$\psi^*(v) = \max_{c \in \text{dom}\,\psi} \left\{ c^\top v - \psi(c) \right\} = - \min_{c \in \text{dom}\,\psi} \left\{ \psi(c) - c^\top v \right\}, \tag{8}$$

where $\text{dom}\,\psi$ denotes the domain of $\psi$. Since $\psi^*(v)$ is a point-wise maximum of linear functions in $v$, it is easy to see that $\psi^*$ is convex. We denote the second Fenchel conjugate of $\psi$ by $\psi^{**}$, which for any $c \in \text{dom}\psi$ is given by:

$$\psi^{**}(c) = \max_{v \in \text{dom}\,\psi^*} \left\{ c^\top v - \psi^*(v) \right\} = - \min_{v \in \text{dom}\,\psi^*} \left\{ \psi^*(v) - c^\top v \right\}, \tag{9}$$

where $\text{dom}\,\psi^*$ is the subset of $\mathbb{R}^K$ for which $\psi^*$ is defined. For example, if $\psi$ is monotonically non-decreasing in an argument, $\text{dom}\,\psi^*$ contains vectors that are non-negative in that coordinate. For convex functions $\psi$, $\psi^{**}(c) = \psi(c)$, $\forall c \in \text{dom}\,\psi$.

The subdifferential $\partial\psi(c)$ of a convex function $\psi$ at a point $c \in \mathbb{R}^K$ is the set of all vectors $v \in \mathbb{R}^K$ such that $f(c') \geq f(c) + v^\top(c' - c)$, $\forall c' \in \mathbb{R}^K$. It follows from the definition of the Fenchel conjugate that for any $c \in \text{dom}\,\psi$,

$$v \in \partial\psi(c) \iff \psi(c) + \psi^*(v) = v^\top c. \tag{10}$$

This is referred to as the *Fenchel-Young equality* [51]. A straight-forward consequence of (10) is that for any $c \in \text{dom}\,\psi$,

$$v^* \in \partial\psi(c) \iff v^* \in \underset{v \in \text{dom}\,\psi^*}{\text{argmax}} \left\{ c^\top v - \psi^*(v) \right\}. \tag{11}$$

**Online convex optimization**  Consider an online convex optimization problem with a bounded convex set $\mathcal{C}$, and a sequence of convex functions $h_1, h_2, \ldots, h_T : \mathcal{C} \to \mathbb{R}$. In each round $t$, the learner needs to choose a point $c_t \in \mathcal{C}$ and incurs a loss $h_t(c_t)$. The goal is to minimize the learner's average regret: $\mathcal{R}_T = \frac{1}{T} \sum_{t=1}^{T} h_t(c_t) - \min_{c \in \mathcal{C}} \frac{1}{T} \sum_{t=1}^{T} h_t(c)$.

Of course, the learner could implement a *best response* (BR) strategy, i.e. simply output the minimizer of $h_t$ at each step: $c_t \in \text{argmin}_{c \in \mathcal{C}} \, h_t(c)$. This trivially gives a no-regret guarantee, but would be expensive in practice.

A more practical and simple approach to the above problem is the classical *follow-the-leader* (FTL) algorithm, which simply outputs a point with the least total loss for the previous rounds:

$$c_{t+1} = \text{FTL}_{\mathcal{C}}(h_1, \ldots, h_t) \in \text{argmin}_{c \in \mathcal{C}} \sum_{\tau=1}^{t} h_{\tau}(c).$$

If each $h_t$ is Lipchitz and strongly convex, then FTL enjoys the following no-regret guarantee: $\mathcal{R}_T = \mathcal{O}\left(\frac{\log(T)}{T}\right)$.

Another popular approach is *online gradient descent* (OGD), which performs gradient updates based on the previously chosen point: $c_{t+1} = \Pi_{\mathcal{C}}\left(c_t - \nabla h_t(c_t)\right)$, where $\nabla h_t(c_t) \in \partial h_t(c_t)$ and $\Pi_{\mathcal{C}}$ denotes the $\ell_2$-projection onto $\mathcal{C}$. If each $h_t$ is Lipchitz, then OGD has the following guarantee: $\mathcal{R}_T = \mathcal{O}\left(\sqrt{\frac{1}{T}}\right)$.

# B   Connections to Previous Algorithms

## B.1   Relationship to Existing Algorithms for (P1)

We elaborate on the three previous algorithms listed in Table 2 and show how their updates can be seen as special cases of our approach. We present these methods using the notations used in this paper and re-written to handle a convex function $\psi : [0, 1]^K \to \mathbb{R}$ that is monotonic in each argument. We point out these approaches were not previously derived using the three-player game view-point that we present in this paper.

**SPADE:**   This method [14] seeks to optimize a convex function of two rates: $\psi(R_1(\theta), R_2(\theta))$, and does so by replacing the rates with convex surrogates $\tilde{R}_1$ and $\tilde{R}_2$, and performing gradient updates on $\theta$ and additional dual variables $\alpha, \beta \in \mathbb{R}_+$:

$$\theta^{t+1} \leftarrow \Pi_{\Theta}\left(\theta^t - \eta\,\alpha^t\,\nabla\tilde{R}_1(\theta) - \eta\beta^t\,\nabla\tilde{R}_2(\theta)\right) \tag{12}$$

$$(\alpha^{t+1}, \beta^{t+1}) \leftarrow \Pi_{\mathcal{A}}\left((\alpha^t, \beta^t) + \eta'(\tilde{R}_1(\theta), \tilde{R}_2(\theta)) - \eta'\nabla\psi^*(\alpha^t, \beta^t)\right), \tag{13}$$

where $\mathcal{A} \subseteq \mathbb{R}_+^2$ is a bounded set that contains all gradients of $\psi$ and $\eta, \eta' > 0$. It is straight-forward to show that the above updates can be recovered from our proposed framework by having both the $\theta$- and $\lambda$-players play OGD on the surrogate Lagrangian, and the $\xi$-player play best response.

**Theorem 5.** *With slack variables $\xi_1, \xi_2 \in \mathbb{R}_+$ and Lagrange multipliers $\alpha, \beta \in \mathbb{R}_+$, define $\tilde{\mathcal{L}}(\theta, \xi; \alpha, \beta) = \psi(\xi_1, \xi_2) + \alpha(\tilde{R}_1(\theta) - \xi_1) + \beta(\tilde{R}_2(\theta) - \xi_2)$. Starting with the same $\theta^0, \alpha^0, \beta^0$, the following updates generate the same iterates $\theta^t$ as the updates in (12) and (13):*

$$\xi^t \leftarrow \underset{\xi \in \mathbb{R}_+^2}{\text{argmin}}\ \tilde{\mathcal{L}}(\theta^t, \xi; \alpha^t, \beta^t)$$

$$\theta^{t+1} \leftarrow \Pi_{\Theta}\left(\theta^t - \eta\,\nabla_{\theta}\,\tilde{\mathcal{L}}(\theta^t, \xi^t; \alpha^t, \beta^t)\right)$$

$$(\alpha^{t+1}, \beta^{t+1}) \leftarrow \Pi_{\mathcal{A}}\left((\alpha^t, \beta^t) + \eta'\,\nabla_{\alpha, \beta}\,\tilde{\mathcal{L}}(\theta^t, \xi^t; \alpha^t, \beta^t)\right).$$

The proof is straight-forward and follows from Lemma 1, which shows that a solution to the $\xi$-player's update is $\xi^t = \nabla\psi^*(\alpha^t, \beta^t)$. Substituting this into the expression for the $\lambda$-player's update gives us the same updates as in (12) and (13).

**NEMSIS:**   This method [15] was designed to work with nested convex functions $\psi$, i.e. functions $\psi$ that are a convex function of multiple convex functions. While these nested metrics can be easily handled within our framework by introducing separate slack variables for each of the inner convex functions, here we consider an application of this method to a simpler metric of the form

$\psi(R_1(\theta), R_2(\theta))$. The NEMSIS updates for this simpler metric with a smooth, convex $\psi$ is given by:

$$\theta^{t+1} \leftarrow \Pi_\Theta \left( \theta^t - \eta\, \alpha^t\, \nabla\, \tilde{R}_1(\theta) - \eta\beta^t\, \nabla\, \tilde{R}_2(\theta) \right) \tag{14}$$

$$(\alpha^{t+1}, \beta^{t+1}) \leftarrow \underset{(\alpha,\beta)\in\mathcal{A}}{\mathrm{argmax}} \sum_{\tau=1}^{t} g_\tau(\alpha, \beta)$$

$$\text{where } g_\tau(\alpha, \beta) := \alpha\tilde{R}_1(\theta^\tau) + \beta\tilde{R}_2(\theta^\tau) - \psi^*(\alpha, \beta). \tag{15}$$

These updates can be recovered from our framework by having the $\lambda$-player play FTL on the combined objective that includes the min over $\xi$, and having the $\theta$-player play OGD, with both players operating on the surrogate Lagrangian.

**Theorem 6.** *With slack variables $\xi_1, \xi_2 \in \mathbb{R}_+$ and Lagrange multipliers $\alpha, \beta \in \mathbb{R}_+$, define $\tilde{\mathcal{L}}_2(\theta; \alpha, \beta) = \alpha\tilde{R}_1(\theta) + \beta\tilde{R}_2(\theta)$ and $\tilde{\mathcal{L}}(\theta, \xi; \alpha, \beta) = \psi(\xi_1, \xi_2) - \alpha\xi_1 - \beta\xi_2 + \tilde{\mathcal{L}}_2(\theta; \alpha, \beta)$. Starting with the same $\theta^0, \alpha^0, \beta^0$, the following updates generate the same iterates $\theta^t$ as the updates in (14) and (15):*

$$\theta^{t+1} \leftarrow \Pi_\Theta \left( \theta^t - \eta \nabla_\theta \tilde{\mathcal{L}}_2(\theta^t; \alpha^t, \beta^t) \right)$$

$$(\alpha^{t+1}, \beta^{t+1}) \leftarrow \mathrm{FTL}_\mathcal{A} \left( \left\{ \min_{\xi \in \mathbb{R}_+^2} \tilde{\mathcal{L}}(\theta^\tau, \xi; \cdot, \cdot) \right\}_{\tau=1}^{t} \right)$$

The proof follows in a straight-forward manner from the definition of the Fenchel conjugate of $\psi$ in (8): $\psi^*(\alpha, \beta) = -\min_{\xi_1, \xi_2 \in \mathbb{R}_+} \{ \psi(\xi_1, \xi_2) - \alpha\xi_1 - \beta\xi_2 \}$.

**F-W:** The Frank-Wolfe based method of Narasimhan et al. [20] optimizes performance metrics $\psi(R_1(\theta), \ldots, R_K(\theta))$ that are defined by a smooth, convex function $\psi$. This employs the classical Frank-Wolfe algorithm along with a cost-sensitive oracle to perform the inner linear optimization needed to be solved within the Frank-Wolfe. Specifically, the algorithm maintains iterates $\theta^t \in \Theta$ and $\mathbf{r}^t \in [0,1]^K$. Starting with $\mathbf{r}^0 = \mathbf{R}(\mathbf{0})$, it performs the following updates:

$$\theta^{t+1} \leftarrow \underset{\theta \in \Theta}{\mathrm{argmin}} \sum_{k=1}^{K} v_k^t R_k(\theta), \text{ where } \mathbf{v}^t \leftarrow \nabla\psi(\mathbf{r}^t) \tag{16}$$

$$\mathbf{r}^{t+1} \leftarrow \left( 1 - \frac{1}{t} \right) \mathbf{r}^t + \frac{1}{t} \mathbf{R}(\theta^{t+1}), \tag{17}$$

where the first step uses a cost-sensitive oracle. The output of this algorithm after $T$ iterations is a stochastic model with probability $\frac{1}{t} \prod_{\tau=t+1}^{T} \left( 1 - \frac{1}{\tau} \right)$ on $\theta^t$.

These updates can be recovered from our game formulation by having the $\theta$-player play OGD and the $\lambda$-player play FTL on a combined objective that contains a min over $\xi$, with both players operating on the true Lagrangians.

**Theorem 7.** *With slack variables $\xi \in [0,1]^K$ and Lagrange multipliers $\lambda \in \mathbb{R}_+^K$, define $\mathcal{L}(\theta, \xi; \lambda) = \psi(\xi) + \sum_{k=1}^{K} \lambda_k(R_k(\theta) - \xi_k)$ and $\mathcal{L}_2(\theta; \lambda) = \sum_{k=1}^{K} \lambda_k R_k(\theta)$. Let $\Lambda = \mathbb{R}_+^K$. Then starting with $\lambda^0 = \nabla\psi(\mathbf{R}(\mathbf{0}))$, the following updates generate the same iterates $\theta^t$ as (16) and (17):*

$$\theta^{t+1} \leftarrow \underset{\theta \in \Theta}{\mathrm{argmin}} \, \mathcal{L}_2(\theta; \lambda^t) \tag{18}$$

$$\lambda^{t+1} \leftarrow \mathrm{FTL}_\Lambda \left( \left\{ \min_{\xi \in [0,1]^K} \mathcal{L}(\theta^\tau, \xi; \cdot) \right\}_{\tau=1}^{t} \right) \tag{19}$$

*Proof.* The proof follows directly from a recent result by Abernethy and Wang [43] that shows that the classical Frank-Wolfe algorithm for convex optimization can be viewed as computing the equilibrium of a zero-sum game. We first note that by definition of the Fenchel conjugate of $\psi$:

$$\min_{\xi \in [0,1]^K} \mathcal{L}(\theta, \xi; \lambda) = -\psi^*(\lambda) + \sum_{k=1}^{K} \lambda_k R_k(\theta).$$

Table 5: Convergence guarantees for algorithms for (P1). Frank-Wolfe [20], SPADE [14], NEMSIS [15] are previous algorithms. Algorithms require $\psi$ to be *smooth* or not; apply either to a general convex model class $\Theta$ or to only a *restricted* subset of models for which the surrogates evaluate to values within the domain of $\psi$; may need access to a cost-sensitive *oracle* or not; output either a stochastic classifier $\bar{\mu}$ or a deterministic classifier $\bar{\theta}$. The bounds are on the optimality gap.

| Alg. | Smooth? | Restr. $\Theta$? | Oracle? | Opt. Gap | Opt. Bound |
|------|---------|------------------|---------|----------|------------|
| F-W | ✓ | ✗ | ✓ | $\psi\left(\mathbf{R}(\bar{\mu})\right) - \min_{\mu \in \Delta_\Theta} \psi\left(\mathbf{R}(\mu)\right)$ | $\tilde{\mathcal{O}}\left(\frac{1}{T} + \rho\right)$ |
| SPADE | ✗ | ✓ | ✗ | $\psi\left(\tilde{\mathbf{R}}(\bar{\theta})\right) - \min_{\theta \in \Theta} \psi\left(\tilde{\mathbf{R}}(\theta)\right)$ | $\tilde{\mathcal{O}}\left(\sqrt{\frac{1}{T}}\right)$ |
| NEMSIS | ✓ | ✓ | ✗ | $\psi\left(\tilde{\mathbf{R}}(\bar{\theta})\right) - \min_{\theta \in \Theta} \psi\left(\tilde{\mathbf{R}}(\theta)\right)$ | $\tilde{\mathcal{O}}\left(\sqrt{\frac{1}{T}}\right)$ |
| Alg. 1 | ✗ | ✗ | ✓ | $\psi\left(\mathbf{R}(\bar{\mu})\right) - \min_{\mu \in \Delta_\Theta} \psi\left(\mathbf{R}(\mu)\right)$ | $\tilde{\mathcal{O}}\left(\sqrt{\frac{1}{T}}\right) + \rho$ |
| Alg. 2 | ✗ | ✗ | ✗ | $\psi\left(\mathbf{R}(\bar{\mu})\right) - \min_{\theta \in \tilde{\Theta}} \psi\left(\tilde{\mathbf{R}}(\theta)\right)$ | $\tilde{\mathcal{O}}\left(\sqrt{\frac{1}{T}}\right)$ |

We expand the FTL update and show that $\lambda^t$ here plays the same role as $\mathbf{v}^t$ in the F-W method.

$$
\begin{aligned}
\lambda^{t+1} &= \underset{\lambda \in \mathbb{R}_+^K}{\operatorname{argmax}} \left\{ -t\psi^*(\lambda) + \sum_{\tau=1}^{t}\sum_{k=1}^{K} \lambda_k R_k(\theta^\tau) \right\} \\
&= \underset{\lambda \in \mathbb{R}_+^K}{\operatorname{argmax}} \left\{ -\psi^*(\lambda) + \sum_{k=1}^{K} \lambda_k \left( \frac{1}{t}\sum_{\tau=1}^{t} R_k(\theta^\tau) \right) \right\} \\
&= \nabla\psi\left( \frac{1}{t}\sum_{\tau=1}^{t} R_k(\theta^\tau) \right) = \nabla\psi(\mathbf{r}^t) = \mathbf{v}^t,
\end{aligned}
$$

where the third equality follows from (11). With the above observation, it is clear that (16) and (18) generate the same iterates $\theta^t$. □

See Table 5 for a comparison of the convergence guarantees for the different algorithms for (P1).

### B.2 Relationship to Existing Algorithms for (P2)

In Table 6, we present a comparison of the convergence guarantees for the different algorithms for (P2). A previous oracle-based approach for solving (P2) is COCO [11], which extends the Frank-Wolfe approach for the unconstrained rate metrics [20] with an outer gradient ascent solver, but does not directly fit into our three-player framework. This method enjoys similar convergence guarantees as Algorithm 1, but requires an additional smoothness assumption on $\phi^j$'s and a more complicated set of updates.

Another previous approach that can be adapted to solve (P2) is the SPADE algorithm, originally proposed for unconstrained problems [14] (see Table 2 and 5), where the $\theta$- and $\lambda$-players perform OGD updates on surrogate objectives, and the $\xi$-player plays best response. We call this adaptation of SPADE as SPADE+ and present its convergence guarantee in Table 6 (with further details in the next section). As with existing surrogate-based methods, SPADE+ requires that the surrogate rates do not evaluate to values outside the domain of $\phi^j$'s. Algorithm 2 does not have this restriction.

## C  Connections to Cotter et al. [23]

Like us, Cotter et al. [23] proposed using surrogates *only* for one of the two players of their game, however, unlike us, they did not prove a convergence result for such an approach, when using the usual Lagrangian formulation. Instead, they proposed a different non-zero-sum game, which they

Table 6: Convergence guarantees for algorithms for (P2). COCO [11] is a previous algorithm and SPADE+ is an adaptation of SPADE [14] to constrained problems. Algorithms apply either to a general convex model class $\Theta$ or to a *restricted* subset for which the surrogates evaluate to values within the domain of $\phi^j$'s; require access to a cost-sensitive *oracle* or not; output a *stochastic* classifier $\bar{\mu}$ or a deterministic classifier $\bar{\theta}$. We denote $g(\bar{\mu}) = \mathbf{E}_{\theta \sim \bar{\mu}}[g(\theta)]$ and $\omega \in (0, 1/2)$. COCO requires $\phi^j$'s to be smooth. The bounds are on the optimality gap and on the maximum constraint violation: $\max_j \phi^j(\mathbf{R}(\cdot))$.

| Alg. | Restr. $\Theta$? | Oracle? | Opt. Gap | | Opt. Bound | Max. Viol. |
|---|---|---|---|---|---|---|
| COCO | ✗ | ✓ | $g(\bar{\mu}) - \min_{\substack{\theta \in \Theta \\ \phi^j(\mathbf{R}(\theta)) \leq 0, \forall j}} g(\theta)$ | | $\tilde{\mathcal{O}}\left(\sqrt{\frac{1}{T}}\right) + \rho$ | $\tilde{\mathcal{O}}\left(\sqrt{\frac{1}{T}}\right) + \rho$ |
| SPADE+ | ✓ | ✗ | $g(\bar{\theta}) - \min_{\substack{\theta \in \Theta \\ \phi^j(\tilde{\mathbf{R}}(\theta)) \leq 0, \forall j}} g(\theta)$ | | $\tilde{\mathcal{O}}\left(\sqrt{\frac{1}{T}}\right)$ | $\tilde{\mathcal{O}}\left(\sqrt{\frac{1}{T}}\right)$ |
| Alg. 1 | ✗ | ✓ | $g(\bar{\mu}) - \min_{\substack{\mu \in \Delta_\Theta \\ \phi^j(\mathbf{R}(\mu)) \leq 0, \forall j}} g(\mu)$ | | $\tilde{\mathcal{O}}\left(\sqrt{\frac{1}{T}} + \rho\right)$ | $\tilde{\mathcal{O}}\left(\sqrt{\frac{1}{T}} + \rho\right)$ |
| Alg. 2 | ✗ | ✗ | $g(\bar{\mu}) - \min_{\substack{\theta \in \tilde{\Theta} \\ \phi^j(\tilde{\mathbf{R}}(\theta)) \leq 0, \forall j}} g(\theta)$ | | $\tilde{\mathcal{O}}\left(\frac{1}{T^{1/2-\omega}}\right)$ | $\tilde{\mathcal{O}}\left(\frac{1}{T^{\omega}}\right)$ |

called the proxy-Lagrangian. In the context of (P1), we can write the proxy-Lagrangian of (2) as:

$$\mathcal{L}_{\theta,\xi}(\theta, \xi; \lambda) = \lambda_1 \psi(\xi_1, \ldots, \xi_K) + \sum_{k=1}^{K} \lambda_{k+1}\left(\tilde{R}_K(\theta) - \xi_k\right)$$

$$\mathcal{L}_{\lambda}(\theta, \xi; \lambda) = \sum_{k=1}^{K} \lambda_{k+1}(R_K(\theta) - \xi_k)$$

where $\lambda \in \Delta^{K-1} \subseteq \mathbb{R}^K$ is the $(K-1)$-probability-simplex—notice that, unlike for the Lagrangian formulation, a multiplier $\lambda_1$ is associated with the *objective*, in addition to the $K$ multipliers associated with the constraints. A similar variant could be easily given for (P2). The only difference from their presentation is that we have grouped $\theta$ and $\xi$ together as the parameters of the first player. Cotter et al. [23]'s proposed approach to such a problem is to minimize $\mathcal{L}_{\theta,\xi}$ over $\theta$ and $\xi$ using an external regret minimizing algorithm (e.g. OGD), and to maximize $\mathcal{L}_\lambda$ over $\lambda$ using a *swap regret* minimizing algorithm. They go on to prove a $\mathcal{O}(1/\sqrt{T})$ convergence rate.

This discussion shows that our proposed approach can be plugged-in to theirs straightforwardly. The resulting algorithm is more complicated (due to the use of swap regret), but the convergence guarantee has a better $T$-dependence. Unfortunately, since they do not use the Lagrangian formulation, and their $\lambda$-player minimizes swap regret, it's difficult to see how analytically optimizing over $\xi$, as we do in Algorithms 1 and 2, could be incorporated into their approach; one can instead perform OGD on $\xi$.

## D  Limitations with Existing Surrogate-based Methods

Existing methods for solving (P1) either assume access to an idealized oracle that can optimize a linear combination of the rates $R_1, \ldots, R_K$ [18–22], or resort to optimizing a convex relaxation to the performance metric [14, 15]. The latter class of methods enjoy convergence guarantees under more realistic assumptions, and proceed by replacing the rates $R_k$ with continuous and differentiable surrogate functions $\tilde{R}_k : \Theta \to \mathbb{R}_+$, so that the resulting objective $\psi(\tilde{R}_1(\theta), \ldots, \tilde{R}_K(\theta))$ is a convex (or pseudo-convex) upper-bound on $\psi(R_1(\theta), \ldots, R_K(\theta))$. However, finding such global convex relaxations to (P1) may not be possible for many common performance metrics.

For example, if $\psi$ is convex but monotonically decreasing in $R_k$, then one would need to replace $R_k$ with a surrogate $\tilde{R}_k$ that is concave in $\theta$ and lower bounds $R_k$: $R_k(\theta) \geq \tilde{R}_k(\theta), \forall \theta \in \Theta$. This can be problematic for metrics such as the G-mean $\psi(z, z') = 1 - \sqrt{z_1 z_2}$, where any non-trivial concave

Figure 2: Left: Plot of negative logarithm as a function of rate. Right: A hinge-based concave lower bound on the indicator function.

surrogates that lower bounds the TPR and TNR will necessarily evaluate to negative values for some $\theta$'s, rendering the square root undefined. A similar issue arises with the KLD and F-measure metrics. While one can still apply existing surrogate methods to these metrics by restricting the model space $\Theta$ to only those $\theta$'s for which the surrogates are non-negative, this severely limits the class of models for which these methods are guaranteed to converge.

Surrogate relaxations pose further complications when used to approximate constraints (P2), where we run the risk of over-constraining the model space, and making the problem infeasible.

## D.1 The Case of the KL-divergence

We take the example of the KL-divergence metric in Table 1 and explain why it is difficult to apply previous surrogate-based methods to optimize this metric, and why our framework provides a cleaner solution.

**Difficulty with SPADE & NEMSIS.** Consider the negative logarithm of the prediction rate of a model: $-\log(\mathbf{E}_X(\mathbb{I}\{f_\theta(X) \geq 0\}))$. This is one of the terms in the KL-divergence metric in Table 1. Previous surrogate methods such as SPADE [14] or NEMSIS [15] optimize this metric by first replacing the indicator function within the logarithm with a surrogate function such as the hinge-loss to obtain a convex upper bound, and then using this surrogate approximation for all updates. The only issue with this approach is that the log is not defined for negative values, and as we shall see below, any reasonable choice of surrogate to replace the inner indicator would produce values that are negative.

To see this, first note that $-\log$ is convex, but non-increasing in its input. Hence to construct a convex upper bound for the negative log-rate, we would have to replace the inner indicator with a concave lower-bounding surrogate. However, if we insist that the surrogate needs to be both concave and lower-bound the indicator function, then we would have to allow the surrogate to be negative for some parts of the input space (unless the surrogate is a constant function). For example, a hinge-based convex upper bound on the negative log metric would look like $-\log(\mathbf{E}_X(\min\{1, f_\theta(X)\}))$. But if $f_\theta(X)$ takes large negative values for a large-enough portion of the input space $X$, the term within the log would be negative, rendering this function ill-defined. See Figure 2 for an illustration.

The above issue points to a drawback with the SPADE and NEMSIS methods that rely heavily on the use of surrogates for optimization. For instance, to optimize $-\log(\mathbf{E}_X(\min\{1, f_\theta(X)\}))$ using the NEMSIS method and a hinge-based surrogate [15], we would implement the following dual update (see (15)):

$$\alpha^{t+1} \leftarrow \operatorname*{argmax}_{\alpha \in \mathbb{R}_+} \left\{ \log(\alpha + \epsilon) - \alpha \frac{1}{t} \sum_{\tau=1}^{t} \mathbf{E}_X \left[\min\{1, f_{\theta^\tau}(X)\}\right] \right\}.$$

A small $\epsilon > 0$ is added to the log argument to avoid numerical issues. Note that if the term $\mathbf{E}_X[\min\{1, f_{\theta^\tau}(X)\}]$ is negative at any point during the course of optimization, the maximization over $\alpha$ becomes unbounded, and the iterates will never converge from that point onwards.

One way NEMSIS can be applied to handle log-rates, is to strongly regularize the model to not output large negative values. This approach may however unnecessarily restrict the space of models we are allowed to optimize over, and may prevent us from finding good solutions.

**Applying Algorithm 2 to KLD.** On the other hand, Algorithm 2 offers a cleaner solution to dealing with log-rates and performance measures based on the KL-divergence. By using original rates (instead of surrogates) for the updates on $\lambda$, the proposed algorithm ensures that the game play between $\lambda$ and $\xi$ never produces values that are outside the domain of $\psi$.

For completeness, we derive the updates for Algorithm 2 for minimizing the KL-divergence metric between the true positive proportion $p$ and the model's prediction rate $\hat{p}(\theta)$. The optimization problem (with constants removed) is given by:

$$\min_{\theta \in \Theta} -p \log(\hat{p}(\theta)) - (1-p) \log(1 - \hat{p}(\theta)).$$

Introducing slack variables for the terms within the log, this can be re-written as:

$$\min_{\theta \in \Theta, \, \xi \in [0,1]^2} -p \log(\xi_1) - (1-p) \log(\xi_2) \quad \text{s.t.} \quad \xi_1 \leq \hat{p}(\theta), \;\; \xi_2 \leq 1 - \hat{p}(\theta). \tag{20}$$

Further, formulating the Lagrangian:

$$\mathcal{L}(\theta, \xi; \lambda) = -p \log(\xi_1) - (1-p) \log(\xi_2) + \lambda_1(\xi_1 - \hat{p}(\theta)) + \lambda_2(\xi_2 - (1 - \hat{p}(\theta))),$$

we seek to solve the following max-min optimization problem:

$$\max_{\lambda \in \mathbb{R}_+^2} \min_{\theta \in \Theta, \, \xi \in [0,1]^2} \mathcal{L}(\theta, \xi; \lambda).$$

Notice that the optimal Lagrange multipliers $\lambda$ are always non-negative.

Using a concave lower-bounding surrogate $\tilde{p}_+$ for $\hat{p}$ and a concave lower-bounding surrogate $\tilde{p}_-$ for $1 - \hat{p}$, we have the following updates for Algorithm 2:

$$\xi_1^{t+1} = \frac{p}{\lambda_1^t + \epsilon}; \;\; \xi_2^{t+1} = \frac{1-p}{\lambda_2^t + \epsilon}$$

$$\theta^{t+1} = \Pi_\Theta \left( \theta^t - \eta_\theta \left( \lambda_1^t \, \nabla_\theta \left[ -\tilde{p}_+(\theta^t) \right] + \lambda_2^t \, \nabla_\theta \left[ -\tilde{p}_-(\theta^t) \right] \right) \right)$$

$$\lambda^{t+1} = \Pi_\Lambda \left( \lambda^t + \eta_\lambda \left( \xi^{t+1} - \begin{bmatrix} \hat{p}(\theta^t) \\ 1 - \hat{p}(\theta^t) \end{bmatrix} \right) \right)$$

where $\Lambda \subset \mathbb{R}_+$ is a bounded set and $\epsilon > 0$ is a very small value to avoid numerical issues. Because $\lambda \geq \mathbf{0}$, both $\xi_1$ and $\xi_2$ are always non-negative, ensuring that the $\log$ in (20) is always defined.

**Other Metrics.** As noted earlier, the above issue also arises with the G-mean metric, where the square root is undefined for negative values, as well as fractional-linear metrics such as the F-measure. In the case of fractional-linear metrics, one can obtain a pseudo-convex upper bound for the metric by replacing the numerator with a convex upper-bounding surrogate and the denominator with a concave lower-bounding surrogate; however the pseudo-convex property of the resulting function holds only if the surrogate for the denominator evaluates to non-negative values, which as with the KLD, poses a restriction on the model class we are allowed to use.

# E   Precision-Recall AUC and Related Metrics

Eban et al. [10] proposed maximizing the area under the precision-recall curve (PR-AUC) using a Riemann approximation. We'll begin by describing their approach, for which the first step is to define a recall@precision$(p; f)$ primitive:

$$\text{recall@precision}(p; f) = \text{recall}(f - t) \;\; \text{s.t.} \;\; \text{precision}(f - t) \geq p$$

Notice that this primitive has not only a value that depends on an implicit threshold $t$ (i.e. recall$(f-t)$), but also a constraint that defines $t$ (i.e. precision$(f - t) \geq p$). We can write it more explicitly by substituting the definitions of precision and recall:

$$\text{recall@precision}(p; f) = \frac{\#\text{TP}(f - t)}{\#\text{LP}} \;\; \text{s.t.} \;\; \frac{\#\text{TP}(f - t)}{\#\text{TP}(f - t) + \#\text{FP}(f - t)} \geq p$$

$$= \frac{\#\text{TP}(f - t)}{\#\text{LP}} \;\; \text{s.t.} \;\; (1-p)\,\#\text{TP}(f - t) \geq p\,\#\text{FP}(f - t)$$

where $\#\mathrm{TP}(f - t)$ is the number of true positive predictions made by classifier $f$ at threshold $t$, $\#\mathrm{FP}(f - t)$ is the number of false positives, and $\#\mathrm{LP}$ is the number of positively-labeled examples (which is a constant independent of $f$).

For a given number of "bins" $b$, we then take a Riemann approximation of PR-AUC by defining $p_i = (2i - 1)/2b$ for $i \in \{1, \ldots, b\}$ to be the bin "centers" along the precision axis, and writing:

$$\mathrm{PR\text{-}AUC}(f) \approx \frac{1}{b} \sum_{i=1}^{b} \mathrm{recall@precision}(p_i; f) \tag{21}$$

Plugging in the definition of $\mathrm{recall@precision}(p_i; f)$ shows that maximizing PR-AUC can be approximated as the constrained optimization problem:

$$\max_{f,t} \ \frac{1}{b\#\mathrm{LP}} \sum_{i=1}^{b} \#\mathrm{TP}(f - t_i)$$

$$\text{s.t. } \forall i \in \{1, 2, \ldots, b\} \, . \, (1 - p_i) \, \#\mathrm{TP}(f - t_i) \geq p_i \#\mathrm{FP}(f - t_i)$$

This problem has the significant advantage of being linear in the counts $\#\mathrm{TP}(f - t_i)$ and $\#\mathrm{FP}(f - t_i)$, and Eban et al. [10] show that it can work well in practice. However, precision, unlike recall, cannot necessarily *achieve* all values in $[0, 1]$ as one varies the threshold. In other words, some of the constraints of this problem may necessarily be *infeasible*. For this reason, we'd instead like to take the Riemann sum along the *recall* axis, instead of the precision axis.

### E.1 Aside: inverse-precision@recall

Eban et al. [10] observed that, if one wishes to maximize precision subject to a recall constraint, then one can equivalently *minimize* $1/\text{precision}$ subject to the same constraint:

$$\min_f \ 1 + \frac{\#\mathrm{FP}(f)}{\#\mathrm{TP}(f)}$$

$$\text{s.t. } \frac{\#\mathrm{TP}(f)}{\#\mathrm{LP}} \geq r$$

Assuming that the constraint will hold with equality at the optimum, one can substitute $\#\mathrm{TP}(f) = r\#\mathrm{LP}$ into the objective function, yielding:

$$\min_f \ 1 + \frac{\#\mathrm{FP}(f)}{r\#\mathrm{LP}}$$

$$\text{s.t. } \frac{\#\mathrm{TP}(f)}{\#\mathrm{LP}} \geq r$$

As desired, both the objective and constraint are now linear combinations of the counts.

### E.2 Solution: precision@recall

Remember that we're interested in taking the Riemann approximation over the recall axis:

$$\mathrm{PR\text{-}AUC}(f) \approx \frac{1}{b} \sum_{i=1}^{b} \mathrm{precision@recall}(r_i; f)$$

for $r_i = (2i - 1)/2b$. Writing $\mathrm{precision@recall}(r_i; f)$ in terms of counts:

$$\mathrm{precision@recall}(r; f) = \frac{\#\mathrm{TP}(f - t)}{\#\mathrm{TP}(f - t) + \#\mathrm{FP}(f - t)} \ \text{s.t. } \frac{\#\mathrm{TP}(f - t)}{\#\mathrm{LP}} \geq r$$

The main problem here is that the value is a ratio, and we cannot immediately apply Eban et al. [10]'s inverse-precision trick since we ultimately want to average over the precisions themselves. However, when combined with our proposed approach, the same trick *does* work. Substituting $\#\mathrm{TP}(f) = r\#\mathrm{LP}$ into the objective yields that:

$$\mathrm{precision@recall}(r; f) = \frac{1}{1 + \frac{\#\mathrm{FP}(f - t)}{r\#\mathrm{LP}}} \ \text{s.t. } \frac{\#\mathrm{TP}(f - t)}{\#\mathrm{LP}} \geq r$$

---

**Algorithm 4** SPADE+ for (P2)

---

Initialize: $\theta^0$, $\lambda^0$

**for** $t = 0$ to $T - 1$ **do**

    $\xi^t \in \text{argmin}_{\xi \in [0,1]^K} \mathcal{L}_1(\xi; \lambda^t)$

    $\theta^{t+1} \leftarrow \Pi_\Theta(\theta^t - \eta_\theta \nabla_\theta \tilde{\mathcal{L}}_2(\theta^t; \lambda^t))$

    $\lambda^{t+1} \leftarrow \Pi_\Lambda(\lambda^t + \eta_\lambda \nabla_\lambda \tilde{\mathcal{L}}(\xi^t, \theta^t; \lambda^t))$

**end for**

**return** $\bar{\theta} = \frac{1}{T} \sum_{t=1}^T \theta^t$

---

We now introduce a slack variable $\gamma$:

$$\text{precision@recall}(r; f) = \frac{1}{1 + \gamma/r}$$

$$\text{s.t.} \quad \frac{\#\text{TP}(f - t)}{\#\text{LP}} \geq r$$

$$\frac{\#\text{FP}(f - t)}{\#\text{LP}} \geq \gamma$$

The value is convex in $\gamma \geq 0$, and the constraints are linear in the counts. Hence, we can approximately maximize PR-AUC with the following constrained optimization problem:

$$\max_{f, t, \gamma \geq 0} \frac{1}{b} \sum_{i=1}^b \frac{1}{1 + \gamma_i/r_i}$$

$$\text{s.t.} \; \forall i \in \{1, 2, \ldots, b\} \, . \, \frac{\#\text{TP}(f - t_i)}{\#\text{LP}} \geq r_i$$

$$\forall i \in \{1, 2, \ldots, b\} \, . \, \frac{\#\text{FP}(f - t_i)}{\#\text{LP}} \geq \gamma_i$$

Since the recall can take on any value in $[0, 1]$ as we vary the threshold, the constraints of this formulation, unlike that of (21), are always feasible.

## F  SPADE+: An Adaptation of SPADE for (P2)

As an additional comparison in our KLD fairness experiments in Section 5, we extend the SPADE algorithm of Narasimhan et al. [14], originally proposed for unconstrained optimization of generalized rate metrics, to the constrained problem in (P2). We seek to solve following max-min problem, where we use a convex surrogate relaxation to the Lagrangian for both the $\theta$- and $\lambda$-player:

$$\max_{\lambda \in \mathbb{R}_+^K} \min_{\substack{\theta \in \Theta, \\ \xi \in [0,1]^K}} \underbrace{\mathcal{L}_1(\xi; \lambda) + \tilde{\mathcal{L}}_2(\theta; \lambda)}_{\tilde{\mathcal{L}}(\xi, \theta; \lambda)}, \tag{22}$$

The procedure is outlined in Algorithm 4, where the $\theta$- and $\lambda$-players perform OGD updates on surrogate objectives, and the $\xi$-player plays best response. Since the max-min objective in (22) is convex in $\theta$, the algorithm returns the average of the model across all iterates. We can then show the following convergence guarantee for Algorithm 4 under the assumption that $\Theta$ only contains models for which each $\phi^j(\tilde{\mathbf{R}}(\theta))$ is defined.

**Theorem 8.** *Let $\bar{\theta}$ be the model returned by Algorithm 4. Let $\Theta$ be a bounded convex set such that $\tilde{\mathbf{R}}(\theta) \in [0, 1]^K$ for all $\theta \in \Theta$. Let $\theta^* \in \Theta$ be such that $\theta^*$ is feasible, i.e. $\phi^j(\tilde{\mathbf{R}}(\theta^*)) \leq 0$, $\forall j \in [J]$, and $g(\theta^*) \leq g(\theta)$ for all $\theta \in \Theta$ that are feasible. Suppose there exists a $\theta' \in \Theta$ such that $\phi^j(\tilde{\mathbf{R}}(\theta')) \leq -\gamma$, $\forall j \in [J]$, for some $\gamma > 0$. Let $B_g = \max_{\theta \in \Theta} g(\theta)$. Let $B_\Theta \geq \max_{\theta \in \Theta} \|\theta\|_2$, $B_\theta \geq \max_t \|\nabla_\theta \tilde{\mathcal{L}}_2(\theta^t; \lambda^t)\|_2$ and $B_\lambda \geq \max_t \|\nabla_\lambda \tilde{\mathcal{L}}(\xi^t, \theta^t; \lambda^t)\|_2$. Then setting $\kappa = 2(L + 1)B_g/\gamma$, $\eta_\theta = \frac{B_\Theta}{B_\theta \sqrt{2T}}$ and $\eta_\lambda = \frac{\kappa}{B_\lambda \sqrt{2T}}$, we have w.p. $\geq 1 - \delta$ over draws of stochastic gradients:*

$$g(\bar{\theta}) \leq g(\theta^*) + \mathcal{O}\left(\sqrt{\frac{\log(1/\delta)}{T}}\right) \quad \text{and} \quad \phi^j(\tilde{\mathbf{R}}(\bar{\theta})) \leq \mathcal{O}\left(\sqrt{\frac{\log(1/\delta)}{T}}\right), \quad \forall j \in [J].$$

The proof follows from an adaptation of the proofs for Theorems 3 and 4 (see Sections J and K). The iterates of SPADE+ form an approximate mixed Nash equilibrium of the zero-sum game in (22).

Table 7: Datasets used in our experiments.

| Dataset | No. of instances | No. of features | Protected Attribute |
|---------|-----------------|----------------|--------------------|
| COMPAS | 4073 | 31 | Gender |
| Communities & Crime | 1495 | 135 | Race Percentage |
| Law School | 15388 | 36 | Race |
| Adult | 32561 | 122 | Gender |
| WikiToxicity | 95692 | 100 | Term 'Gay' |

## G  Additional Experimental Details

**Datasets.** A summary of the datasets is provided in Table 7. Wiki Toxicity is a text dataset, and we use the Glove embedding [49] to convert the text to numerical features.

**Implementation Details.** We implemented Algorithms 2–3 using the open-source Tensorflow Constrained Optimization (TFCO) library[3] of Cotter et al. (2019) [23, 24]. This library will soon provide direct support for optimizing several ratio-based metrics (using the approach presented in Algorithm 3), as well as, the the precision-recall and PR-AUC metrics described in Appendix E.

All comparisons use a linear model. We use hinge loss based surrogates $\tilde{R}_k$ for the rates. We use Adam to perform full gradient updates on $\theta$ and $\lambda$ and run our algorithms for a total of 5000 iterations. The datasets are split randomly into train-validation-test sets in the ratio 4/9:2/9:1/3, except WikiToxicity where we use the splits made available by the authors [47].

**Hyperparameter Choices.** The proposed algorithm uses Adam for all gradient updates and is run for 5000 iterations. The step-sizes for the $\lambda$- and $\theta$-updates in the proposed algorithms are chosen from the range $\{0.001, 0.01, 0.1, 1.0\}$ using the validation set. We use a heuristic provided in Cotter et al. (2019) [24] to pick the hyperparameters that best trade-off between the fairness objective and constraint violations. We record snapshots of the iterates of our algorithms every 10 iterations and construct both a stochastic classifier and a deterministic classifier from the iterates.

**Shrinking.** The final stochastic classifier in all our theoretical results is defined by a uniform distribution over $T$ deterministic classifiers. The large support size may make this stochastic classifier undesirable in practice. In our experiments, we post-process the iterates of our algorithms to construct a sparse stochastic classifier over only $J + 1$ iterates (where recall $J$ is the number of constraints in (P2)). Specifically, we adopt the shrinking procedure of Cotter et al. (2019) [24] and solve the following linear program over the $T$-dimensional simplex:

$$\min_{\mu \in \mathbb{R}_+^T, \; \sum_{t=1}^T \mu_t = 1} \sum_{t=1}^T \mu_t \, g(\theta^t) \;\; \text{s.t.} \;\; \sum_{t=1}^T \mu_t \, \phi^j(\theta^t) \leq 0, \;\; \forall j \in [J]. \tag{23}$$

We then use the optimal weighting $\mu^*$ for this problem to construct the final stochastic classifier. Since (23) is a linear optimization over the simplex with $J$ linear constraints, the solution $\mu^*$ can be shown to have at most $J + 1$ non-zero entries [24]. Further, when the constraint functions $\phi^j$'s are convex, this solution is also feasible for our original constrained problem (P2), i.e. $\sum_{t=1}^T \mu_t^* \, \phi^j \left( \mathbf{R}(\theta^t) \right) \leq 0 \Rightarrow \phi^j \left( \mathbf{E}_{\theta \sim \mu^*}[\mathbf{R}(\theta)] \right) \leq 0, \forall j \in [J]$. Thus in this case final stochastic classifier constructed from (23) is both sparse and feasible. Even when the constraint functions are non-convex in the rates, we find the shrinking procedure to be often effective in producing sparse stochastic classifiers that are feasible on the training set.

### G.1  Additional Comparisons for KL-divergence Experiments

We continue with the first task presented in Section 5 on optimizing the KLD fairness metric subject to error rate constraints. We first provide additional details on the methods we compare against. Algorithm 2 uses Adam for the $\theta$ and $\lambda$ updates, and computes the best response for the $\xi$-player using an analytical closed-form solution. We implement UncError by optimizing the hinge loss. Both UncError and the logistic regression for COCO and PostShift are trained with 2500 iterations of Adam. COCO is run for 500 outer iterations and 10 inner iterations. The step-size parameter for these

Table 8: Optimizing KL-divergence fairness metric s.t. error rate constraints. For each method, we report two metrics: A (B), where A is the test fairness metric (*lower* is better) and B is the ratio of the test error rate of the method and that of a classifier that optimizes unconstrained error rate (*lower* is better). During training, we constrain B to be $\leq 1.1$. Among the last 4 columns, the lowest fairness metric is highlighted in blue, and the second-lowest is shown in light blue.

| | UncError | PostShift | SPADE+ | COCO | Algorithm 2 | |
| | | | | | Stochastic | Determ. |
|---|---|---|---|---|---|---|
| COMPAS | 0.115 (1.00) | 0.000 (1.01) | 0.009 (1.03) | 0.043 (1.01) | 0.000 (1.03) | 0.000 (1.03) |
| Crime | 0.224 (1.00) | 0.005 (1.40) | 0.185 (0.82) | 0.252 (0.83) | 0.120 (1.11) | 0.146 (1.08) |
| Law | 0.199 (1.00) | 0.001 (1.45) | 0.040 (1.09) | 0.043 (1.05) | 0.054 (1.12) | 0.056 (1.08) |
| Adult | 0.114 (1.00) | 0.000 (1.22) | 0.071 (1.03) | 0.011 (1.10) | 0.014 (1.10) | 0.014 (1.10) |
| Wiki | 0.175 (1.00) | 0.001 (1.21) | 0.083 (1.18) | 0.134 (1.17) | 0.133 (1.09) | 0.127 (1.18) |

Table 9: Same as Table 3, except we compare SPADE+, COCO and Algorithm 2 *without* the post-processing shrinking procedure in (23) to construct the final stochastic classifier. SPADE+ outputs $\frac{1}{T} \sum_{t=1}^{T} \theta^t$. COCO outputs a stochastic classifier specified by the weighting scheme provided in Narasimhan (2018) [11]. Algorithm 2 outputs a stochastic classifier specified by a uniform distribution over all iterates.

| | SPADE+ | COCO | Algorithm 2 |
|---|---|---|---|
| COMPAS | 0.000 (1.02) | 0.000 (1.07) | 0.016 (1.02) |
| Crime | 0.170 (0.83) | 0.042 (1.14) | 0.188 (1.00) |
| Law | 0.296 (1.01) | 0.027 (1.13) | 0.074 (1.11) |
| Adult | 0.098 (1.01) | 0.003 (1.31) | 0.068 (1.08) |
| Wiki | 0.106 (1.23) | 0.010 (1.39) | 0.098 (1.30) |

three methods is chosen from $\{0.005, 0.01, 0.05, \ldots, 10.0\}$. SPADE+ uses Adam for both the $\theta$ and $\lambda$ updates, is run for 5000 iterations, with the two step-sizes chosen from $\{0.001, 0.01, 0.1, 1.0, 10.0\}$. For a fair comparison, we apply the post-processing shrinking step both to our method and to COCO, i.e. we apply (23) to their iterates and construct stochastic classifiers.

We include one additional method for comparison: the surrogate-based method SPADE+ described in Appendix F, an adaptation of the SPADE method [14] to constrained problems. Here again apply shrinking to construct a stochastic classifier from the final iterates. We evaluate all methods based on their (a) KLD fairness metric on the test set, and (b) their constraint violation on the test set, measured by the ratio of error rate of the learned model and that of the unconstrained model: $e\hat{r}r(f_\theta)/e\hat{r}r(f_{unc})$. The results, with SPADE+ include, are shown in Table 8. SPADE+ yields significantly poor fairness values on the Crime and Adult datasets and suffers high constraint violation on the Wiki dataset. In contrast, the stochastic classifier trained by Algorithm 2 closely satisfies the error rate constraint on almost all datasets. Also on four datasets, the proposed algorithm achieves the best or second-best fairness metric, doing significantly better than SPADE+ and COCO on the Crime dataset.

All three constrained optimization algorithms benefit from using the shrinking procedure to post-process their iterates and construct the final stochastic classifier. To better analyze their convergence behavior, we also report in Table 9, their performance without the shrinking procedure. Specifically, for each method, we construct the final classifier as prescribed by its convergence gurantee: for SPADE+, this a deterministic classifier given by the average model parameters: $\sum_{t=1}^{T} \theta^t$ (see Theorem 8); for COCO, this is a stochastic classifier specified by the weighting scheme provided in Narasimhan [11]; for Algorithm 2, this is a stochastic classifier specified by a uniform distribution over all iterates (see Theorem 4).

Without the post-processing step, SPADE+ tends to overconstrain the model, and yields relatively poor fairness objective on the Law, Adult and Wiki datasets. This is because of its heavy dependence on surrogates. In contrast, even without the shrinking step, Algorithm 2 closely satisfies the constraint on four of five datasets, while suffering an increase in the fairness metric. COCO suffers a higher constraint violation on all datasets in the absence of the shrinking step.

# H   Proof of Theorem 1

We first note that while the definition of the cost-sensitive oracle in Definition 1 assumes a deterministic guarantee, Theorem 1 easily extend to the case where the guarantee on the oracle holds with high probability. For example, our results easily apply to an oracle that minimizes the empirical cost-sensitive error on a training sample $S = \{(x_1, y_1), \ldots, (x_n, y_n)\}$ over the space of models $\Theta$. In this case, using standard uniform convergence arguments for a bounded $\Theta$, one can show that for any $\lambda$, with probability $\geq 1 - \delta$ (over draw of $S$), the empirical error minimizer $\theta^*$ satisfies

$\mathcal{L}_2(\theta^*; \lambda) \leq \min_{\theta \in \Theta} \mathcal{L}_2(\theta; \lambda) + \mathcal{O}\left(\sqrt{\frac{\log(1/\delta)}{n}}\right).$

Before moving to the proof of Theorem 1, we first state a lemma that relates the Lagrangian $\mathcal{L}$ with the Fenchel conjugate of $\psi$. In particular, we will show that if $\psi$ is $L$-Lipschitz, it suffices to set the radius of the space of Lagrange multipliers over which we optimize $\mathcal{L}$ to be at most $L$. This will later be helpful in choosing $\kappa$ in Algorithm 1.

**Lemma 2.** *Suppose $\psi$ is convex, monotonically non-decreasing in each argument and $L$-Lipschitz w.r.t. the $\ell_\infty$-norm. Setting the radius of the space of Lagrange multipliers $\Lambda$ to be at most $L$, i.e. $\Lambda = \{\lambda \in \mathbb{R}_+^K : \|\lambda\|_1 \leq L\}$, we have the following for any $\mu \in \Delta_\Theta$.*

*1.* $\lambda^* \in \operatorname*{argmax}_{\lambda \in \mathbb{R}_+^K} \left\{ \min_{\xi \in [0,1]^K} \mathcal{L}(\xi, \mu; \lambda) \right\} \Rightarrow \lambda^* \in \Lambda.$

*2.* $\psi(\mathbf{R}(\mu)) = \max_{\lambda \in \Lambda} \left\{ \min_{\xi \in [0,1]^K} \mathcal{L}(\xi, \mu; \lambda) \right\}.$

*3. if additionally, $\psi$ is strictly convex with $\psi(\mathbf{0}) = \mathbf{0}$,*

$$\nabla \psi^*(\lambda^*) \in \operatorname*{argmin}_{\xi \in [0,1]^K} \mathcal{L}_1(\xi; \lambda^*), \ \forall \lambda^* \in \Lambda.$$

*Proof.* 1. By strong duality, we first have:

$$\psi(\mathbf{R}(\mu)) = \max_{\lambda \in \mathbb{R}_+^K} \min_{\xi \in [0,1]^K} \mathcal{L}(\xi, \mu; \lambda) \tag{24}$$

From the Fenchel-Young's equality in (10), we have:

$$\lambda^* = \nabla \psi(\mathbf{R}(\mu)) \iff \psi(\mathbf{R}(\mu)) = -\psi^*(\lambda^*) + \sum_{k=1}^K \lambda_k^* R_k(\mu)$$

$$\iff \psi(\mathbf{R}(\mu)) = \min_{\xi \in [0,1]^K} \left\{ \psi(\xi) - \sum_{k=1}^K \lambda_k^* \xi_k \right\} + \sum_{k=1}^K \lambda_k^* R_k(\mu)$$

$$\iff \psi(\mathbf{R}(\mu)) = \min_{\xi \in [0,1]^K} \mathcal{L}(\xi, \mu; \lambda^*)$$

$$\iff \max_{\lambda \in \mathbb{R}_+^K} \min_{\xi \in [0,1]^K} \mathcal{L}(\xi, \mu; \lambda) = \min_{\xi \in [0,1]^K} \mathcal{L}(\xi, \mu; \lambda^*)$$

$$\iff \lambda^* \in \operatorname*{argmax}_{\lambda \in \mathbb{R}_+^K} \min_{\xi \in [0,1]^K} \mathcal{L}(\xi, \mu; \lambda),$$

where where the second step uses the definition of Fenchel dual $\psi^*$ (see (8)) and the fact that domain of $\psi$ is $[0,1]^K$, and the fourth step follows from (24). uses monotonicity of $\psi$. and the fact the gradients of $\psi$ all have non-zero entries. This shows that $\nabla \psi(\mathbf{R}(\mu))$ is a maximizer of $\min_\xi \mathcal{L}(\xi, \mu; \lambda)$ over $\lambda \in \mathbb{R}_+^K$. Because $\psi$ is $L$-Lipschitz w.r.t. the $\ell_\infty$-norm and $\mathbf{R}(\mu) \in [0,1]^K$, the gradient norm $\|\nabla \psi(\mathbf{R}(\mu))\|_1 \leq L$. Hence, the set $\Lambda$ always contains a maximizer of $\min_\xi \mathcal{L}(\xi, \mu; \lambda)$ over $\lambda \in \mathbb{R}_+^K$.

2. For any $\mu$:

$$\max_{\lambda \in \Lambda} \min_{\xi \in [0,1]^K} \mathcal{L}(\xi, \mu; \lambda) = \max_{\lambda \in \Lambda} \left\{ \min_{\xi \in [0,1]^K} \left\{ \psi(\xi) - \sum_{k=1}^K \lambda_k \xi_k \right\} + \sum_{k=1}^K \lambda_k R_k(\mu) \right\}$$

$$= \max_{\lambda \in \Lambda} \left\{ -\psi^*(\lambda) + \sum_{k=1}^{K} \lambda_k R_k(\mu) \right\}$$

$$= \max_{\lambda \in \mathbb{R}_+^K} \left\{ -\psi^*(\lambda) + \sum_{k=1}^{K} \lambda_k R_k(\mu) \right\}$$

$$= \psi^{**}(\mathbf{R}(\mu))$$

$$= \psi(\mathbf{R}(\mu)),$$

where in the fourth step, we have used statement 1 to replace the max of $\Lambda$ with a max over $\mathbb{R}_+^K$; the fifth step uses the definition of second Fenchel conjugate (see (9)); the last step follows from convexity of $\psi$.

3. Because $\psi$ is strictly convex, $\nabla \psi(\mathbf{0}) = \mathbf{0}$, and (by Lipschitzness) the largest gradient of $\psi$ has an $\ell_1$-norm $L$, it follows that every $\lambda \in \Lambda$ is sub-gradient of $\psi$ at some point in $[0,1]^K$. The strict convexity of $\psi$ also gives us that $\psi^*$ is differentiable at any $\lambda \in \Lambda$ (see e.g. [51]). We then have from the Fenchel-Young's equality in (10):

$$\xi^* = \nabla \psi^*(\lambda^*) \quad \Longleftrightarrow \quad \psi^*(\lambda^*) = \psi(\xi^*) - \sum_{k=1}^{K} \lambda_k^* \xi_k^*$$

$$\Longleftrightarrow \quad \xi^* \in \operatorname*{argmin}_{\xi \in [0,1]^K} \left\{ \psi(\xi) - \sum_{k=1}^{K} \lambda_k^* \xi_k \right\} = \operatorname*{argmin}_{\xi \in [0,1]^K} \mathcal{L}_1(\xi; \lambda^*).$$

This completes the proof of parts 1-3. $\qquad \square$

## H.1   Proof of Lemma 1

The proof follows directly from statement 3 of Lemma 2.

## H.2   General Convergence Result

We present a convergence result for a general no-regret strategy for the $\lambda$-player, and then apply it to the case where the player runs OGD with specific step-sizes. The iterates generated by Algorithm 1 for (P1) yield an approximate Nash equilibrium, i.e. the $\lambda$-player choosing the fixed strategy $\bar{\lambda} = \frac{1}{T} \sum_{t=1}^{T} \lambda^t$, the $\xi$-player choosing the fixed strategy $\bar{\xi} = \frac{1}{T} \sum_{t=1}^{T} \xi^t$, and the $\theta$-player choosing a uniform distribution $\bar{\mu}$ over $\theta^1, \ldots, \theta^T$, together comprise an approximate mixed-strategy Nash equilibrium of the zero-sum game in (3).

**Theorem 9.** *Let* $\theta^1, \ldots, \theta^T, \xi^1, \ldots, \xi^T, \lambda^1, \ldots, \lambda^T$ *be the iterates generated by Algorithm 1 for* (P1). *Suppose the iterates satisfy the following:*

$$\mathcal{L}_1(\xi^t; \lambda^t) = \min_{\xi \in [0,1]^K} \mathcal{L}_1(\xi; \lambda^t) \tag{25}$$

$$\mathcal{L}_2(\theta^t; \lambda^t) \leq \min_{\theta \in \Theta} \mathcal{L}_2(\theta; \lambda^t) + \rho \tag{26}$$

$$\frac{1}{T} \sum_{t=1}^{T} \mathcal{L}(\xi^t, \theta^t; \lambda^t) \geq \max_{\lambda \in \Lambda} \frac{1}{T} \sum_{t=1}^{T} \mathcal{L}(\xi^t, \theta^t; \lambda) - \epsilon_\lambda, \tag{27}$$

*for some* $\epsilon_\lambda > 0$. *Suppose* $\psi$ *is convex, monotonically non-decreasing in each argument and $L$-Lispchitz w.r.t. $\ell_\infty$ norm. Let $\bar{\mu}$ be a stochastic model with a probability mass of $\frac{1}{T}$ on $\theta^t$. Then setting $\kappa = L$:*

$$\psi\big(\mathbf{R}(\bar{\mu})\big) \leq \min_{\mu \in \Delta_\Theta} \psi\big(\mathbf{R}(\mu)\big) + \epsilon_\lambda + \rho.$$

*Proof.* From (25) and (26), we have:

$$\frac{1}{T} \sum_{t=1}^{T} \mathcal{L}(\xi^t, \theta^t; \lambda^t) = \frac{1}{T} \sum_{t=1}^{T} \min_{\xi \in [0,1]^K} \mathcal{L}(\xi, \theta^t; \lambda^t)$$

$$\leq \frac{1}{T}\sum_{t=1}^{T}\min_{\xi\in[0,1]^K,\theta\in\Theta}\mathcal{L}(\xi,\theta;\lambda^t) + \rho$$

$$= \frac{1}{T}\sum_{t=1}^{T}\min_{\xi\in[0,1]^K,\mu\in\Delta_\Theta}\mathcal{L}(\xi,\mu;\lambda^t) + \rho \quad \text{(by linearity of } \mathcal{L} \text{ in } \mu\text{)}$$

$$\leq \min_{\xi\in[0,1]^K,\mu\in\Delta_\Theta}\frac{1}{T}\sum_{t=1}^{T}\mathcal{L}(\xi,\mu;\lambda^t) + \rho$$

$$= \min_{\xi\in[0,1]^K,\mu\in\Delta_\Theta}\mathcal{L}\left(\xi,\mu;\bar{\lambda}\right) + \rho \quad \text{(by linearity of } \mathcal{L} \text{ in } \lambda^t\text{)}$$

$$\leq \max_{\lambda\in\Lambda}\min_{\xi\in[0,1]^K,\mu\in\Delta_\Theta}\mathcal{L}(\xi,\mu;\lambda) + \rho$$

$$= \min_{\mu\in\Delta_\Theta}\left\{\max_{\lambda\in\Lambda}\min_{\xi\in[0,1]^K}\mathcal{L}(\xi,\mu;\lambda)\right\} + \rho$$

$$= \min_{\mu\in\Delta_\Theta}\psi(\mathbf{R}(\mu)) + \rho, \tag{28}$$

where in the second-last step we interchange the max and min using the fact that $\mathcal{L}$ is linear in $\mu$, linear in $\lambda$ and convex in $\xi$; the last step follows from statement 2 of Lemma 2 (given that the radius of the space of Lagrange multipliers $\kappa$ is set to $L$).

Next, from the theorem statement, the OGD updates on $\lambda$ satisfy:

$$\frac{1}{T}\sum_{t=1}^{T}\mathcal{L}(\xi^t,\theta^t;\lambda^t) \geq \max_{\lambda\in\Lambda}\frac{1}{T}\sum_{t=1}^{T}\mathcal{L}(\xi^t,\theta^t;\lambda) - \epsilon_\lambda$$

$$\geq \min_{\xi\in[0,1]^K}\max_{\lambda\in\Lambda}\frac{1}{T}\sum_{t=1}^{T}\mathcal{L}(\xi,\theta^t;\lambda) - \epsilon_\lambda$$

$$\geq \max_{\lambda\in\Lambda}\min_{\xi\in[0,1]^K}\frac{1}{T}\sum_{t=1}^{T}\mathcal{L}(\xi,\theta^t;\lambda) - \epsilon_\lambda$$

$$= \max_{\lambda\in\Lambda}\min_{\xi\in[0,1]^K}\mathcal{L}(\xi,\bar{\mu};\lambda) - \epsilon_\lambda$$

$$= \psi(\mathbf{R}(\bar{\mu})) - \epsilon_\lambda, \tag{29}$$

where in the third step, we interchange the min and max using the fact that $\mathcal{L}$ is convex in $\xi$ and linear in $\lambda$; the last step follows from statement 2 of Lemma 2.

Combining (28) and (29), we have the desired result. $\qquad\square$

### H.3 Corollary for OGD on $\lambda$

*Proof of Theorem 1.* We apply standard OGD convergence analysis [41] to the $\lambda$-player's gradient updates on $\lambda$ and standard arguments to convert a uniform regret guarantee into a stochastic one [50] (see e.g. Theorem 7 in Cotter et al. (2019) [23]). For the sequence of losses $-\mathcal{L}(\xi^1,\theta^1;\cdot),\ldots,-\mathcal{L}(\xi^T,\theta^T;\cdot)$, with $\eta = \frac{\kappa}{B_\lambda\sqrt{2T}}$ in Algorithm 1, we get the following regret bound. With probability at least $1 - \delta$ over draws of stochastic gradients of $\mathcal{L}$:

$$\frac{1}{T}\sum_{t=1}^{T}\mathcal{L}(\xi^t,\theta^t;\lambda^t) \geq \max_{\lambda\in\Lambda}\frac{1}{T}\sum_{t=1}^{T}\mathcal{L}(\xi^t,\theta^t;\lambda) - 2\kappa B_\lambda\sqrt{\frac{1 + 16\log(1/\delta)}{T}}.$$

For the above, we use the fact that the gradients $\nabla_\lambda\mathcal{L}(\theta^t,\xi^t;\lambda^t)$ are unbiased, i.e. $\mathbf{E}\left[\nabla_\lambda\mathcal{L}(\theta^t,\xi^t;\lambda^t)\right] \in \partial_\lambda\mathcal{L}(\theta^t,\xi^t;\lambda^t)$.

Following statement 1 in Lemma 2, we set the radius of the space of Lagrange multipliers $\Lambda$ to $\kappa = L$ and apply Theorem 9 (along with Lemma 1 for the $\xi$-player's best response, and the CSO oracle assumption for the $\theta$-player's best response) to complete the proof. We get with probability at least $1 - \delta$ over draws of the stochastic gradients:

$$\psi\left(\mathbf{R}(\bar{\mu})\right) \leq \min_{\mu\in\Delta_\Theta}\psi\left(\mathbf{R}(\mu)\right) + 2L B_\lambda\sqrt{\frac{1 + 16\log(1/\delta)}{T}} + \rho.$$

$\square$

# I Proof of Theorem 2

We first state a lemma that relates the surrogate Lagrangian with the Fenchel conjugate of $\psi$:

**Lemma 3.** *Let* $\tilde{\mathcal{L}}(\xi, \mu; \lambda) = \mathcal{L}_1(\xi; \lambda) + \tilde{\mathcal{L}}_2(\mu; \lambda)$. *For any* $\theta \in \Theta$, *for which* $\tilde{\mathbf{R}}(\theta) \in [0,1]^K$:

$$\psi(\tilde{\mathbf{R}}(\theta)) = \max_{\lambda \in \mathbb{R}_+^K} \min_{\xi \in [0,1]^K} \tilde{\mathcal{L}}(\xi, \theta; \lambda).$$

*Proof.* The proof follows the same steps as statement 2 of Lemma 2. $\square$

## I.1 General Convergence Result

We present a convergence result for general no-regret strategies for the $\theta$- and $\lambda$-player that reach an approximate coarse-correlated equilibrium, and then apply it to the case where the players run OGD with specific step-sizes.

**Theorem 10.** *Let* $\theta^1, \dots, \theta^T, \xi^1, \dots, \xi^T, \lambda^1, \dots, \lambda^T$ *be the iterates generated by Algorithm 2 for* (P1). *Let* $\tilde{\Theta} = \{\theta \in \Theta \,|\, \tilde{\mathbf{R}}(\theta) \in [0,1]^K\}$. *Suppose the iterates comprise the following approximate coarse-correlated equilibrium:*

$$\frac{1}{T}\sum_{t=1}^{T} \mathcal{L}_1(\xi^t; \lambda^t) \leq \min_{\xi \in [0,1]^K} \frac{1}{T}\sum_{t=1}^{T} \mathcal{L}_1(\xi; \lambda^t); \tag{30}$$

$$\frac{1}{T}\sum_{t=1}^{T} \tilde{\mathcal{L}}_2(\theta^t; \lambda^t) \leq \min_{\theta \in \Theta} \frac{1}{T}\sum_{t=1}^{T} \tilde{\mathcal{L}}_2(\theta; \lambda^t) + \epsilon_\theta; \tag{31}$$

$$\frac{1}{T}\sum_{t=1}^{T} \mathcal{L}(\xi^t, \theta^t; \lambda^t) \geq \max_{\lambda \in \Lambda} \frac{1}{T}\sum_{t=1}^{T} \mathcal{L}(\xi^t, \theta^t; \lambda) - \epsilon_\lambda, \tag{32}$$

*for some* $\epsilon_\theta > 0$ *and* $\epsilon_\lambda > 0$. *Suppose* $\psi$ *is convex, monotonically non-decreasing in each argument and* $L$-*Lispchitz w.r.t.* $\ell_\infty$ *norm. Let* $\bar{\mu}$ *be a stochastic model with a probability mass of* $\frac{1}{T}$ *on* $\theta^t$. *Then setting* $\kappa = L$:

$$\psi(\mathbf{R}(\bar{\mu})) \leq \min_{\theta \in \tilde{\Theta}} \psi(\tilde{\mathbf{R}}(\theta)) + \epsilon_\theta + \epsilon_\lambda$$

*Proof.* We have:

$$
\begin{aligned}
\frac{1}{T}\sum_{t=1}^{T}\tilde{\mathcal{L}}(\xi^t,\theta^t;\lambda^t) &\leq \min_{\xi\in[0,1]^K}\frac{1}{T}\sum_{t=1}^{T}\mathcal{L}_1(\xi;\lambda^t) + \frac{1}{T}\sum_{t=1}^{T}\tilde{\mathcal{L}}_2(\theta^t;\lambda^t) \quad \text{(from (30))}\\
&\leq \min_{\xi\in[0,1]^K}\frac{1}{T}\sum_{t=1}^{T}\mathcal{L}_1(\xi;\lambda^t) + \min_{\theta\in\Theta}\frac{1}{T}\sum_{t=1}^{T}\tilde{\mathcal{L}}_2(\theta;\lambda^t) + \epsilon_\theta \quad \text{(from (31))}\\
&= \min_{\xi\in[0,1]^K,\theta\in\Theta}\frac{1}{T}\sum_{t=1}^{T}\tilde{\mathcal{L}}(\xi,\theta;\lambda^t) + \epsilon_\theta\\
&= \min_{\xi\in[0,1]^K,\theta\in\Theta}\tilde{\mathcal{L}}(\xi,\theta;\bar{\lambda}) + \epsilon_\theta \quad \text{(by linearity of } \tilde{\mathcal{L}} \text{ in } \lambda^t\text{)}\\
&\leq \max_{\lambda\in\mathbb{R}_+^K}\min_{\xi\in[0,1]^K,\theta\in\Theta}\tilde{\mathcal{L}}(\xi,\theta;\lambda) + \epsilon_\theta\\
&\leq \max_{\lambda\in\mathbb{R}_+^K}\min_{\xi\in[0,1]^K,\theta\in\tilde{\Theta}}\tilde{\mathcal{L}}(\xi,\theta;\lambda) + \epsilon_\theta \quad \text{(as } \tilde{\Theta}\subseteq\Theta\text{)}\\
&= \min_{\theta\in\tilde{\Theta}}\left\{\max_{\lambda\in\mathbb{R}_+^K}\min_{\xi\in[0,1]^K}\tilde{\mathcal{L}}(\xi,\theta;\lambda)\right\} + \epsilon_\theta
\end{aligned}
$$

$$= \min_{\theta \in \tilde{\Theta}} \psi(\tilde{\mathbf{R}}(\theta)) + \epsilon_\theta, \tag{33}$$

where in the second-last step, we interchange the max and min using the fact that $\tilde{\mathcal{L}}$ is convex in $\theta$ and $\xi$ and, linear in $\lambda$; the last step follows from Lemma 3.

Next, the OGD updates on $\lambda$ satisfy:

$$
\begin{aligned}
\frac{1}{T} \sum_{t=1}^{T} \mathcal{L}(\xi^t, \theta^t; \lambda^t) &\geq \max_{\lambda \in \Lambda} \frac{1}{T} \sum_{t=1}^{T} \mathcal{L}(\xi^t, \theta^t; \lambda) - \epsilon_\lambda \quad \text{(from (32))} \\
&\geq \min_{\xi \in [0,1]^K} \max_{\lambda \in \Lambda} \frac{1}{T} \sum_{t=1}^{T} \mathcal{L}(\xi, \theta^t; \lambda) - \epsilon_\lambda \\
&= \max_{\lambda \in \Lambda} \min_{\xi \in [0,1]^K} \frac{1}{T} \sum_{t=1}^{T} \mathcal{L}(\xi, \theta^t; \lambda) - \epsilon_\lambda \\
&= \max_{\lambda \in \Lambda} \min_{\xi \in [0,1]^K} \mathcal{L}(\xi, \bar{\mu}; \lambda) - \epsilon_\lambda \\
&= \psi(\mathbf{R}(\bar{\mu})) - \epsilon_\lambda, \tag{34}
\end{aligned}
$$

where in the third step, we interchange the min and max using the fact that $\mathcal{L}$ is convex in $\xi$ and linear in $\lambda$; the last step follows from statement 2 in Lemma 2.

Combining (33) and (34) using the surrogate upper-bounding property, i.e. using $\tilde{\mathcal{L}}(\xi^t, \theta^t; \lambda^t) \geq \mathcal{L}(\xi^t, \theta^t; \lambda^t)$, gives us the desired result.

$\square$

### I.2   Corollary for OGD on $\theta$ and $\lambda$

*Proof of Theorem 2.*  We show that the iterates of Algorithm 2 form an approximate coarse-correlated equilibrium and apply Theorem 10.

From Lemma 1, the best-response strategy of the $\xi$-player gives us:

$$\frac{1}{T} \sum_{t=1}^{T} \mathcal{L}_1(\xi^t; \lambda^t) = \frac{1}{T} \sum_{t=1}^{T} \min_{\xi \in [0,1]^K} \mathcal{L}_1(\xi; \lambda^t) \leq \min_{\xi \in [0,1]^K} \frac{1}{T} \sum_{t=1}^{T} \mathcal{L}_1(\xi; \lambda^t).$$

We then derive no-regret guarantees for the updates of the $\theta$- and $\lambda$-player using standard convergence analysis for OGD [41] and standard online-to-stochastic conversion arguments [50] (see e.g. Theorem 7 in Cotter et al. (2019) [23]). For the sequence of losses $-\mathcal{L}(\xi^1, \cdot; \lambda^1), \ldots, -\mathcal{L}(\xi^T, \cdot; \lambda^T)$ optimized by the $\theta$-player, setting $\eta = \frac{B_\Theta}{B_\theta \sqrt{2T}}$ in Algorithm 2, we get the following regret bound for the $\theta$-player (see Corollary 3 in [23] for the complete derivation). With probability $\geq 1 - \delta/2$ over draws of stochastic gradients of $\tilde{\mathcal{L}}_2$:

$$\frac{1}{T} \sum_{t=1}^{T} \tilde{\mathcal{L}}(\xi^t, \theta^t; \lambda^t) \leq \min_{\theta \in \Theta} \frac{1}{T} \sum_{t=1}^{T} \tilde{\mathcal{L}}(\xi^t, \theta; \lambda^t) + 2 B_\Theta B_\theta \sqrt{\frac{1 + 16 \log(2/\delta)}{T}}.$$

The regret guarantee uses the fact that the gradients $\nabla_\theta \tilde{\mathcal{L}}_2(\theta^t; \lambda^t)$ are unbiased, i.e. $\mathbf{E}\left[\nabla_\theta \tilde{\mathcal{L}}_2(\theta^t; \lambda^t)\right] \in \partial_\theta \tilde{\mathcal{L}}_2(\theta^t; \lambda^t)$.

Similarly, for the sequence of losses $-\mathcal{L}(\xi^1, \theta^1; \cdot), \ldots, -\mathcal{L}(\xi^T, \theta^T; \cdot)$ optimized by the $\lambda$-player, setting $\eta = \frac{\kappa}{B_\lambda \sqrt{2T}}$ in Algorithm 2, we get the following regret bound. With probability $\geq 1 - \delta/2$ over draws of stochastic gradients of $\mathcal{L}$:

$$\frac{1}{T} \sum_{t=1}^{T} \mathcal{L}(\xi^t, \theta^t; \lambda^t) \geq \max_{\lambda \in \Lambda} \frac{1}{T} \sum_{t=1}^{T} \mathcal{L}(\xi^t, \theta^t; \lambda) - 2\kappa B_\lambda \sqrt{\frac{1 + 16 \log(2/\delta)}{T}}.$$

This again uses the fact that the gradients $\nabla_\lambda \mathcal{L}(\theta^t, \xi^t; \lambda^t)$ are unbiased, i.e. $\mathbf{E}\left[\nabla_\lambda \mathcal{L}(\theta^t, \xi^t; \lambda^t)\right] \in \partial_\lambda \mathcal{L}(\theta^t, \xi^t; \lambda^t)$.

Following statement 1 of Lemma 2, we set the radius of the space of Lagrange multipliers $\Lambda$ to $\kappa = L$ and apply Theorem 10 (along with the CSO oracle assumption for the $\theta$-player's best response) to complete the proof. We get with probability $\geq 1 - \delta$ over draws of stochastic gradients of $\mathcal{L}$ and $\tilde{\mathcal{L}}_2$:

$$\psi\big(\mathbf{R}(\bar{\mu})\big) \leq \min_{\theta \in \tilde{\Theta}} \psi\big(\tilde{\mathbf{R}}(\theta)\big) + 2B_\Theta\, B_\theta \sqrt{\frac{1 + 16\log(2/\delta)}{T}} + 2L\, B_\lambda \sqrt{\frac{1 + 16\log(2/\delta)}{T}},$$

as desired. $\square$

## J Proof of Theorem 3

The proof of Theorem 3 adapts ideas from previous results on constrained optimization and game equilibrium [11, 24]. We will find it useful to first prove a couple of lemmas.

**Lemma 4.** *Suppose each $\phi^j$ is convex and monotonically non-decreasing in each argument and $g$ is convex. Let $\mathcal{L}$ be as defined in (5). Let $\mu^* \in \Delta_\Theta$ be such that $\mu^*$ is feasible, i.e. $\phi^j(\mathbf{R}(\mu^*)) \leq 0, \forall j \in [J]$, and $\mathbf{E}_{\theta \sim \mu^*}[g(\theta)] \leq \mathbf{E}_{\theta \sim \mu}[g(\theta)]$ for every $\mu \in \Delta_\Theta$ that is feasible. Further, for $\lambda \in \mathbb{R}_+^{J+K}$, denote $a = [\lambda_1, \ldots, \lambda_J]^\top$ and $b = [\lambda_{J+1}, \ldots, \lambda_{J+K}]^\top$, and for $a \in \mathbb{R}_+^J$, let $\Phi_a(\xi) = \sum_{j=1}^J a_j\, \phi^j(\xi)$. Then for any $\mu \in \Delta_\Theta$ and $a \in \mathbb{R}_+^J$:*

1. $\mathbf{E}_{\theta \sim \mu^*}[g(\theta)] = \max_{\lambda \in \mathbb{R}_+^{J+K}} \min_{\xi \in [0,1]^K,\, \mu \in \Delta_\Theta} \mathcal{L}(\xi, \mu; \lambda).$

2. $\nabla \Phi_a(\mathbf{R}(\mu)) = \operatorname*{argmax}_{b \in \mathbb{R}_+^K} \min_{\xi \in [0,1]^K} \mathcal{L}(\xi, \mu; a, b)$

*Proof.* 1. We re-state $\mathcal{L}$ from (5) for a stochastic classifier $\mu$:

$$\mathcal{L}(\mu, \xi; \lambda) = \mathbf{E}_{\theta \sim \mu}[g(\theta)] + \sum_{j=1}^J \lambda_j\, \phi^j(\xi) + \sum_{k=1}^K \lambda_{J+k}\, (R_k(\mu) - \xi_k).$$

Since $\mathcal{L}$ is linear in $\mu(\theta)$, convex in $\xi$ and linear in $\lambda$, strong duality holds, and we have:

$$\max_{\lambda \in \mathbb{R}_+^{J+K}} \min_{\xi \in [0,1]^K,\, \mu \in \Delta_\Theta} \mathcal{L}(\xi, \mu; \lambda) = \min_{\mu,\xi:\, \xi \geq \mathbf{R}(\mu),\, \phi^j(\xi) \leq 0,\, \forall j} \mathbf{E}_{\theta \sim \mu}[g(\theta)]$$

$$= \min_{\mu:\, \phi^j(\mathbf{R}(\mu)) \leq 0,\, \forall j} \mathbf{E}_{\theta \sim \mu}[g(\theta)] \quad \text{(by monotonicity of } \phi^j\text{'s)}$$

$$= \mathbf{E}_{\theta \sim \mu^*}[g(\theta)].$$

2. Denoting the first $J$ indices of $\lambda$ by $a$ and the remaining indices by $b$, we have:

$$\mathcal{L}(\xi, \mu; a, b) = \mathbf{E}_{\theta \sim \mu}[g(\theta)] + \sum_{j=1}^J a_j\, \phi^j(\xi) + \sum_{k=1}^K b_k\, (R_k(\mu) - \xi_k)$$

By strong duality, for fixed $a \in \mathbb{R}_+^J$:

$$\max_{b \in \mathbb{R}_+^K} \min_{\xi \in [0,1]^K} \mathcal{L}(\xi, \mu; a, b) = \mathbf{E}_{\theta \sim \mu}[g(\theta)] + \sum_{j=1}^J a_j\, \phi^j(\mathbf{R}(\mu))$$

$$= \mathbf{E}_{\theta \sim \mu}[g(\theta)] + \Phi_a(\mathbf{R}(\mu)). \tag{35}$$

From the Fenchel-Young's equality in (10):

$b^* = \nabla \Phi_a(\mathbf{R}(\mu))$

$$\iff \mathbf{E}_{\theta \sim \mu}[g(\theta)] + \Phi_a(\mathbf{R}(\mu)) = \mathbf{E}_{\theta \sim \mu}[g(\theta)] + -\Phi_a^*(b^*) + \sum_{k=1}^K \lambda_k^*\, R_k(\mu)$$

$$\iff \mathbf{E}_{\theta \sim \mu}[g(\theta)] + \Phi_a(\mathbf{R}(\mu)) = \mathbf{E}_{\theta \sim \mu}[g(\theta)] + \min_{\xi \in [0,1]^K}\left\{ \Phi_a(\xi) - \sum_{k=1}^K b_k^*\, \xi_k \right\} + \sum_{k=1}^K b_k^*\, R_k(\mu)$$

$$\iff \quad \mathbf{E}_{\theta \sim \mu}\left[g(\theta)\right] + \Phi_a(\mathbf{R}(\mu)) = \min_{\xi \in [0,1]^K} \mathcal{L}(\xi, \mu; a, b^*)$$

$$\iff \quad \max_{b \in \mathbb{R}_+^K} \min_{\xi \in [0,1]^K} \mathcal{L}(\xi, \mu; a, b) = \min_{\xi \in [0,1]^K} \mathcal{L}(\xi, \mu; a, b^*)$$

$$\iff \quad b^* \in \operatorname*{argmax}_{\lambda \in \mathbb{R}_+^K} \min_{\xi \in [0,1]^K} \mathcal{L}(\xi, \mu; a, b),$$

where where the second step uses the definition of Fenchel dual $\psi^*$ (see (8)) and the fact that domain of $\psi$ is $[0,1]^K$, and the fourth step follows from (35).

$\square$

**Lemma 5.** *Let $\phi : \mathbb{R}_+^K \to \mathbb{R}$ be monotonically non-decreasing in each argument and be L-Lipschitz in w.r.t. $\ell_\infty$ norm. Then for any $\bar{\mu} \in \Delta_\Theta$ and $\bar{\xi} \in \mathbb{R}_+^K$:*

$$\phi(\mathbf{R}(\bar{\mu})) \leq \phi(\bar{\xi}) + L \max_{k \in [K]} (R_k(\bar{\mu}) - \bar{\xi}_k)_+,$$

*where $(z)_+ = \max\{0, z\}$.*

*Proof.* We first show that for any $\xi \in [0,1]^K$ and $\xi + \Delta \in [0,1]^K$, $\phi(\xi + \Delta) - \phi(\xi) \leq L \max_{k \in [K]} (\Delta_k)_+$, and the lemma directly follows from this. Define $\Delta_k^+ = (\Delta_k)_+$ and $\Delta_k^- = (-\Delta_k)_+$. Then

$$
\begin{aligned}
\phi(\xi + \Delta) - \phi(\xi) &= \phi(\xi + \Delta^+ - \Delta^-) - \phi(\xi) \\
&= \phi(\xi + \Delta^+ - \Delta^-) - \phi(\xi - \Delta^-) + \phi(\xi - \Delta^-) - \phi(\xi) \\
&\leq L \max_{k \in [K]} |\Delta_k^+| + \phi(\xi - \Delta^-) - \phi(\xi) \quad \text{(from the Lipchitz property of } \phi\text{)} \\
&\leq L \max_{k \in [K]} |\Delta_k^+| + 0 \quad \text{(from monotonicity of } \phi\text{)} \\
&= L \max_{k \in [K]} (\Delta_k)_+,
\end{aligned}
$$

as desired.

$\square$

## J.1 General Convergence Result

We present a convergence result for a general no-regret strategy for the $\lambda$-player, and then apply it to the case where the player runs OGD with specific step-sizes. In this case, the iterates generated by Algorithm 1 for (P2) yield an approximate Nash equilibrium, i.e. the $\lambda$-player choosing the fixed strategy $\bar{\lambda} = \frac{1}{T} \sum_{t=1}^T \lambda^t$, the $\xi$-player choosing the fixed strategy $\bar{\xi} = \frac{1}{T} \sum_{t=1}^T \xi^t$, and the $\theta$-player choosing a uniform distribution $\bar{\mu}$ over $\theta^1, \ldots, \theta^T$, together form an approximate mixed-strategy Nash equilibrium of the zero-sum game in (6).

**Theorem 11.** *Let $\theta^1, \ldots, \theta^T, \xi^1, \ldots, \xi^T, \lambda^1, \ldots, \lambda^T$ be the iterates generated by Algorithm 1 for (P2) when run with a $\rho$-approximate CSO oracle. Suppose the iterates satisfy the following:*

$$\mathcal{L}_1(\xi^t; \lambda^t) = \min_{\xi \in [0,1]^K} \mathcal{L}_1(\xi; \lambda^t); \tag{36}$$

$$\mathcal{L}_2(\theta^t; \lambda^t) \leq \min_{\theta \in \Theta} \mathcal{L}_2(\theta; \lambda^t) + \rho; \tag{37}$$

$$\frac{1}{T} \sum_{t=1}^T \mathcal{L}(\xi^t, \theta^t; \lambda^t) \geq \max_{\lambda \in \Lambda} \frac{1}{T} \sum_{t=1}^T \mathcal{L}(\xi^t, \theta^t; \lambda) - \epsilon_\lambda, \tag{38}$$

*for some $\rho > 0$ and $\epsilon_\lambda > 0$. Suppose each $\phi^j$ is convex, monotonically non-decreasing in each argument and L-Lispchitz w.r.t. $\ell_\infty$ norm. Suppose there exists a $\mu' \in \Delta_\Theta$ such that $\phi^j(\mathbf{R}(\mu')) \leq -\gamma, \forall j \in [J]$, for some $\gamma > 0$. Let $\bar{\mu}$ be a stochastic model with a probability mass of $\frac{1}{T}$ on $\theta^t$. Let $\mu^* \in \Delta_\Theta$ be such that $\mu^*$ is feasible, i.e. $\phi^j(\mathbf{R}(\mu^*)) \leq 0, \forall j \in [J]$, and $\mathbf{E}_{\theta \sim \mu^*}[g(\theta)] \leq \mathbf{E}_{\theta \sim \mu}[g(\theta)]$ for every $\mu \in \Delta_\Theta$ that is feasible. Let $\lambda^* \in \operatorname{argmax}_{\lambda \in \Lambda} \min_{\xi, \mu} \mathcal{L}(\xi, \mu; \lambda)$. Then setting $\kappa \geq 2\|\lambda^*\|_1$:*

$$\mathbf{E}_{\theta \sim \bar{\mu}}\left[g(\theta)\right] \leq \mathbf{E}_{\theta \sim \mu^*}\left[g(\theta)\right] + \rho + \epsilon_\lambda$$

*and*

$$\phi^j(\mathbf{R}(\bar{\mu})) \leq 2\,(L+1)\,(\rho + \epsilon_\lambda)/\kappa, \quad \forall j \in [J].$$

*Proof.* Let $\bar{\xi} = \frac{1}{T}\sum_{t=1}^{T} \xi^t$. Let $\lambda^*$ be as defined in the theorem statement.

**Optimality.** From (36) and (37), we get:

$$
\begin{aligned}
\frac{1}{T}\sum_{t=1}^{T} \mathcal{L}(\xi^t, \theta^t; \lambda^t) &= \frac{1}{T}\sum_{t=1}^{T} \min_{\xi \in [0,1]^K} \mathcal{L}(\xi, \theta^t; \lambda^t) \\
&\leq \frac{1}{T}\sum_{t=1}^{T} \min_{\xi \in [0,1]^K, \theta \in \Theta} \mathcal{L}(\xi, \theta; \lambda^t) + \rho \\
&= \frac{1}{T}\sum_{t=1}^{T} \min_{\xi \in [0,1]^K, \mu \in \Delta_\Theta} \mathcal{L}(\xi, \mu; \lambda^t) + \rho \quad \text{(by linearity of } \mathcal{L} \text{ in } \mu) \\
&\leq \min_{\xi \in [0,1]^K, \mu \in \Delta_\Theta} \frac{1}{T}\sum_{t=1}^{T} \mathcal{L}(\xi, \mu; \lambda^t) + \rho \\
&= \min_{\xi \in [0,1]^K, \mu \in \Delta_\Theta} \mathcal{L}\left(\xi, \mu; \bar{\lambda}\right) + \rho \quad \text{(by linearity of } \mathcal{L} \text{ in } \lambda^t) \\
&\leq \max_{\lambda \in \mathbb{R}_+^K} \min_{\xi \in [0,1]^K, \mu \in \Delta_\Theta} \mathcal{L}(\xi, \mu; \lambda) + \rho \\
&= \mathbf{E}_{\theta \sim \mu^*}\left[g(\theta)\right] + \rho,
\end{aligned}
\tag{39}
$$

where the last step follows from statement 1 of Lemma 4.

Next, from (38), the OGD updates on $\lambda$ satisfy for any $\lambda' \in \Lambda$:

$$\frac{1}{T}\sum_{t=1}^{T} \mathcal{L}(\xi^t, \theta^t; \lambda^t) \geq \frac{1}{T}\sum_{t=1}^{T} \mathcal{L}(\xi^t, \theta^t; \lambda') - \epsilon_\lambda \tag{40}$$

Combining (39) and (40), we have for any $\lambda' \in \Lambda$:

$$\frac{1}{T}\sum_{t=1}^{T} \mathcal{L}(\xi^t, \theta^t; \lambda') \leq \mathbf{E}_{\theta \sim \mu^*}\left[g(\theta)\right] + \rho + \epsilon_\lambda. \tag{41}$$

Setting $\lambda' = 0$ in (41) gives us:

$$\frac{1}{T}\sum_{t=1}^{T} g(\theta^t) \leq \mathbf{E}_{\theta \sim \mu^*}\left[g(\theta)\right] + \rho + \epsilon_\lambda$$

or

$$\mathbf{E}_{\theta \sim \bar{\mu}}\left[g(\theta)\right] \leq \mathbf{E}_{\theta \sim \mu^*}\left[g(\theta)\right] + \rho + \epsilon_\lambda.$$

This proves the optimality result.

**Feasibility.** Recall that there are two sets of constraints $\phi^j(\mu) \leq 0, \forall j \in [J]$ and $R_k(\mu) \leq \xi_k, k \in [K]$. We first look at the first set of constraints. Let $j' \in \operatorname{argmax}_{j \in [J]} \phi^j(\bar{\xi})$. If we set $\lambda'_{j'} = \lambda^*_{j'} + \kappa/2$ and $\lambda'_j = \lambda^*, \forall j \neq j', j \in [J+K]$ in (41) (note that $\lambda' \in \Lambda$). This gives us:

$$\frac{1}{T}\sum_{t=1}^{T} \mathcal{L}(\xi^t, \theta^t; \lambda^*) + \frac{\kappa}{2T}\sum_{t=1}^{T} \phi^{j'}(\xi^t) \leq \mathbf{E}_{\theta \sim \mu^*}\left[g(\theta)\right] + \rho + \epsilon_\lambda.$$

Since $\mathcal{L}$ is linear in $\mathbf{R}(\theta)$ and convex in $\xi$, using Jensen's inequality, we have $\frac{1}{T}\sum_{t=1}^{T} \mathcal{L}(\xi^t, \theta^t; \lambda^*) \geq \mathcal{L}_1(\bar{\xi}; \lambda^*) + \mathcal{L}_2(\bar{\mu}; \lambda^*) \geq \min_{\xi, \mu} \mathcal{L}(\xi, \mu; \lambda^*) = \mathbf{E}_{\theta \sim \mu^*}\left[g(\theta)\right]$ (by statement 1 of Lemma 4). Further by convexity of $\phi^j$, we have:

$$\mathbf{E}_{\theta \sim \mu^*}\left[g(\theta)\right] + \frac{\kappa}{2}\phi^{j'}(\bar{\xi}) \leq \mathbf{E}_{\theta \sim \mu^*}\left[g(\theta)\right] + \rho + \epsilon_\lambda,$$

which implies:
$$\max_{j\in[J]} \phi^j(\bar{\xi}) \le 2(\rho + \epsilon_\lambda)/\kappa.$$

Applying Lemma 5 to each $\phi^j$, we further get

$$\max_{j\in[J]} \phi^j(\mathbf{R}(\bar{\mu})) \le L \max_{k\in[K]} (R_k(\bar{\mu}) - \bar{\xi}_k)_+ + 2(\rho + \epsilon_\lambda)/\kappa. \qquad (42)$$

For the second set of constraints, let $k' \in \operatorname{argmax}_{k\in[K]}(R_k(\bar{\mu}) - \bar{\xi}_k)$. If $R_{k'}(\bar{\mu}) - \bar{\xi}_{k'} \le 0$, then $\max_{k\in[K]}(R_k(\bar{\mu}) - \bar{\xi}_k)_+ = 0$. Otherwise, set $\lambda'_{J+k'} = \lambda^*_{k'} + \kappa/2$ and $\lambda'_j = \lambda^*, \forall j \ne J + k'$ in (41) in (41), giving us:

$$\frac{1}{T}\sum_{t=1}^{T} \mathcal{L}(\xi^t, \theta^t; \lambda^*) + \frac{\kappa}{2T}\sum_{t=1}^{T}(R_{k'}(\theta^t) - \xi_{k'}^t) \le \mathbf{E}_{\theta\sim\mu^*}[g(\theta)] + \rho + \epsilon_\lambda.$$

Following the same steps as above:
$$R_{k'}(\bar{\mu}) - \bar{\xi}_{k'} \le 2(\rho + \epsilon_\lambda)/\kappa$$

which further gives us:

$$\max_{k\in[K]}(R_k(\bar{\mu}) - \bar{\xi}_k)_+ = \max_{k\in[K]}(R_k(\bar{\mu}) - \bar{\xi}_k) \le 2(\rho + \epsilon_\lambda)/\kappa. \qquad (43)$$

Substituting (43) back in (42), we have:

$$\max_{j\in[J]} \phi^j(\mathbf{R}(\bar{\mu})) \le 2(L+1)(\rho + \epsilon_\lambda)/\kappa,$$

as desired. $\qquad\qquad\qquad\qquad\qquad\qquad\qquad\qquad\qquad\qquad\qquad\qquad\qquad\square$

## J.2 Corollary for OGD on $\lambda$

**Lemma 6.** *Suppose each $\phi^j$ is monotonically non-decreasing in each argument and $L$-Lipschitz w.r.t. the $\ell_\infty$ norm. Suppose there exists a $\mu' \in \Delta_\Theta$ such that $\phi^j(\mathbf{R}(\mu')) \le -\gamma, \forall j \in [J]$, for some $\gamma > 0$. Let $B_g = \max_{\theta\in\Theta} g(\theta)$. Fix $\mu \in \Delta_\Theta$. Then for any*

$$\lambda^* \in \operatorname*{argmax}_{\lambda\in\mathbb{R}_+^{J+K}} \left\{ \min_{\xi\in[0,1]^K, \mu\in\Delta_\Theta} \mathcal{L}(\xi, \mu; \lambda) \right\},$$

*the following holds: $\|\lambda^*\|_1 \le (L+1)B_g/\gamma$.*

*Proof.* We separate $\lambda^*$ into $a^* = [\lambda_1^*, \dots, \lambda_J^*]^\top$ and $b^* = [\lambda_{J+1}^*, \dots, \lambda_{J+K}^*]^\top$. We first bound the norm of $a^*$. Let $\mu^*$ be as defined in Lemma 4. We then have:

$$
\begin{aligned}
\mathbf{E}_{\theta\sim\mu^*}[g(\theta)] &= \min_{\xi\in[0,1]^K, \mu\in\Delta_\Theta} \mathcal{L}(\xi, \mu; \lambda^*) \\
&= \min_{\xi\in[0,1]^K, \mu\in\Delta_\Theta} \mathbf{E}_{\theta\sim\mu}[g(\theta)] + \sum_{j=1}^{J} a_j^* \phi^j(\xi) + \sum_{k=1}^{K} b_k^* (R_k(\mu) - \xi_k) \\
&\le \mathbf{E}_{\theta\sim\mu'}[g(\mu)] + \sum_{j=1}^{J} a_j^* \phi^j(\mathbf{R}(\mu')) \quad (\text{setting } \mu = \mu' \text{ and } \xi_k = R_k(\mu')) \\
&\le B_g - \gamma\sum_{j=1}^{J} a_j^* = B_g - \gamma\|a^*\|_1,
\end{aligned}
$$

which gives us:
$$\|a^*\|_1 \le (B_g - \mathbf{E}_{\theta\sim\mu^*}[g(\theta)])/\gamma \le B_g/\gamma.$$

We next bound the norm of $b^*$. Let $\Phi_{a^*}$ be as defined in Lemma 4. We then have from statement 2 of Lemma 4 that $b^* = \nabla\Phi_{a^*}(\mathbf{R}(\mu^*))$. We further have:

$$\|b^*\|_1 \le \max_{\mu\in\Delta_\Theta} \left\| \nabla\Phi_{a^*}(\mathbf{R}(\mu)) \right\|_1$$

$$
\leq \max_{\xi \in [0,1]^K} \left\| \nabla \Phi_{a^*}(\xi) \right\|_1 \quad \text{(because } \mathbf{R}(\mu) \in [0,1]^K)
$$

$$
= \max_{\xi \in [0,1]^K} \left\| \sum_{j=1}^{J} a_j^* \nabla \phi^j(\xi) \right\|_1
$$

$$
\leq \max_{\xi \in [0,1]^K} \sum_{j=1}^{J} |a_j^*| \left\| \nabla \phi^j(\xi) \right\|_1
$$

$$
\leq \sum_{j=1}^{J} |a_j^*| \max_{\xi \in [0,1]^K} \left\| \nabla \phi^j(\xi) \right\|_1
$$

$$
\leq L \|a^*\|_1 \leq L B_g / \gamma,
$$

where in the last step, we use the Lipschitz property of $\phi^j$.

Thus $\|\lambda^*\|_1 = \|a^*\|_1 + \|b^*\|_1 \leq (L+1) B_g / \gamma$, as desired. $\qquad \square$

*Proof of Theorem 3.* We apply standard OGD convergence analysis [41] and standard online-to-stochastic transformation arguments [50] to the $\lambda$-player's gradient updates on $\lambda$. For the sequence $-\mathcal{L}(\xi^1, \theta^1; \cdot), \ldots, -\mathcal{L}(\xi^T, \theta^T; \cdot)$, with $\eta = \frac{\kappa}{B_\lambda \sqrt{2T}}$ in Algorithm 1, we get the following regret bound. With probability at least $1 - \delta$ over draws of stochastic gradients of $\mathcal{L}$:

$$
\frac{1}{T} \sum_{t=1}^{T} \mathcal{L}(\xi^t, \theta^t; \lambda^t) \geq \max_{\lambda \in \Lambda} \frac{1}{T} \sum_{t=1}^{T} \mathcal{L}(\xi^t, \theta^t; \lambda) - 2\kappa B_\lambda \sqrt{\frac{1 + 16 \log(1/\delta)}{T}}.
$$

Following Lemma 6, we set the radius of the space of Lagrange multipliers $\Lambda$ to $\kappa = 2(L+1)B_g/\gamma$ and apply Theorem 9 to complete the proof. We get with probability at least $1 - \delta$ over draws of the stochastic gradients:

$$
\mathbf{E}_{\theta \sim \bar{\mu}} [g(\theta)] \leq g(\theta^*) + \rho + \frac{4(L+1)B_g B_\lambda}{\gamma} \sqrt{\frac{1 + 16 \log(1/\delta)}{T}}.
$$

and

$$
\max_{j \in [J]} \phi^j(\mathbf{R}(\bar{\mu})) \leq \frac{\gamma \rho}{B_g} + 4(L+1)B_\lambda \sqrt{\frac{1 + 16 \log(1/\delta)}{T}}.
$$

$\qquad \square$

# K  Proof of Theorem 4

The proof adapts ideas from previous results on constrained optimization and game equilibrium [24, 26]. We will find the following lemma useful.

**Lemma 7.** *Let $\tilde{\mathcal{L}}(\xi, \mu; \lambda) = \mathcal{L}_1(\xi; \lambda) + \tilde{\mathcal{L}}_2(\mu; \lambda)$. Suppose each $\phi^j$ is convex and monotonically non-decreasing in its arguments and $g$ is convex. Let $\tilde{\Theta} = \left\{ \theta \in \Theta \,|\, \tilde{\mathbf{R}}(\theta) \in [0,1]^K, \, \forall j \right\}$. Let $\tilde{\theta}^* \in \tilde{\Theta}$ be such that $\phi^j(\tilde{\mathbf{R}}(\tilde{\theta}^*)) \leq 0, \, \forall j \in [J]$ and $g(\tilde{\theta}^*) \leq g(\theta)$ for all $\theta \in \tilde{\Theta}$ that satisfies the same constraints. Then*

$$
\max_{\lambda \in \mathbb{R}_+^K} \min_{\xi \in [0,1]^K, \theta \in \Theta} \tilde{\mathcal{L}}(\xi, \theta; \lambda) \leq g(\tilde{\theta}^*).
$$

*Proof.* Since $\tilde{\mathcal{L}}$ is linear in $\mu(\theta)$, convex in $\xi$ and linear in $\lambda$, strong duality holds and we have:

$$
\max_{\lambda \in \mathbb{R}_+^K} \min_{\xi \in [0,1]^K, \theta \in \Theta} \tilde{\mathcal{L}}(\xi, \mu; \lambda) = \min_{\theta \in \Theta, \, \xi \,|\, \tilde{\mathbf{R}}(\theta) \leq \xi, \, \phi^j(\xi) \leq 0, \, \forall j} g(\theta)
$$

$$
\leq \min_{\theta \in \tilde{\Theta}, \, \xi \,|\, \tilde{\mathbf{R}}(\theta) \leq \xi, \, \phi^j(\xi) \leq 0, \, \forall j} g(\theta) \quad \text{(from } \tilde{\Theta} \subseteq \Theta)
$$

$$
= \min_{\theta \in \tilde{\Theta} \,|\, \phi^j(\tilde{\mathbf{R}}(\theta)) \leq 0, \, \forall j} g(\theta) = g(\tilde{\theta}^*)
$$

where the last step follows from monotonicity of each $\phi^j$ and from the fact that for any $\theta \in \tilde{\Theta}$, the range of $\tilde{\mathbf{R}}(\theta)$ is the same as the constraint set for $\xi$. $\qquad\square$

### K.1 General Convergence Result

We present a convergence result for general no-regret strategies for the $\theta$- and $\lambda$-player that find an approximate coarse-correlated equilibrium, and then specialize it to the case where the players run OGD with specific step-sizes.

**Theorem 12.** *Let $\theta^1, \ldots, \theta^T, \xi^1, \ldots, \xi^T, \lambda^1, \ldots, \lambda^T$ be the iterates generated by Algorithm 2 for* (P2). *Let $\tilde{\Theta} = \{\theta \in \Theta \,|\, \tilde{\mathbf{R}}(\theta) \in [0,1]^K, \forall j\}$. Suppose the iterates comprise an approximate coarse-correlated equilibrium, i.e. satisfy:*

$$\frac{1}{T}\sum_{t=1}^{T}\mathcal{L}_1(\xi^t;\lambda^t) \leq \min_{\xi \in [0,1]^K} \frac{1}{T}\sum_{t=1}^{T}\mathcal{L}_1(\xi;\lambda^t); \tag{44}$$

$$\frac{1}{T}\sum_{t=1}^{T}\tilde{\mathcal{L}}_2(\theta^t;\lambda^t) \leq \min_{\theta \in \Theta} \frac{1}{T}\sum_{t=1}^{T}\tilde{\mathcal{L}}_2(\theta;\lambda^t) + \epsilon_\theta; \tag{45}$$

$$\frac{1}{T}\sum_{t=1}^{T}\mathcal{L}(\xi^t,\theta^t;\lambda^t) \geq \max_{\lambda \in \Lambda} \frac{1}{T}\sum_{t=1}^{T}\mathcal{L}(\xi^t,\theta^t;\lambda) - \epsilon_\lambda, \tag{46}$$

*for some $\epsilon_\theta > 0$ and $\epsilon_\lambda > 0$. Suppose each $\phi^j$ is convex, monotonically non-decreasing in each argument and L-Lispchitz w.r.t. $\ell_\infty$ norm. Let $B_g = \max_{\theta \in \Theta} g(\theta)$. Let $\tilde{\theta}^* \in \tilde{\Theta}$ be such that $\phi^j(\tilde{\mathbf{R}}(\tilde{\theta}^*)) \leq 0, \forall j \in [J]$ and $g(\tilde{\theta}^*) \leq g(\theta)$ for all $\theta \in \tilde{\Theta}$ that satisfies the same constraints. Let $\bar{\mu}$ be a stochastic model with a probability mass of $\frac{1}{T}$ on $\theta^t$. Then:*

$$\mathbf{E}_{\theta \sim \bar{\mu}}\left[g(\theta)\right] \leq g(\tilde{\theta}^*) + \epsilon_\theta + \epsilon_\lambda$$

*and*

$$\phi^j(\mathbf{R}(\bar{\mu})) \leq (L+1)(B_g + \epsilon_\theta + \epsilon_\lambda)/\kappa, \quad \forall j \in [J].$$

*Proof of Theorem 12.* Let $\bar{\xi} = \frac{1}{T}\sum_{t=1}^{T}\xi^t$.

**Optimality.** We have:

$$
\begin{aligned}
\frac{1}{T}\sum_{t=1}^{T}\tilde{\mathcal{L}}(\xi^t,\theta^t;\lambda^t) &\leq \min_{\xi \in [0,1]^K} \frac{1}{T}\sum_{t=1}^{T}\mathcal{L}_1(\xi;\lambda^t) + \frac{1}{T}\sum_{t=1}^{T}\tilde{\mathcal{L}}_2(\theta^t;\lambda^t) && \text{(from (44))} \\
&\leq \min_{\xi \in [0,1]^K} \frac{1}{T}\sum_{t=1}^{T}\mathcal{L}_1(\xi;\lambda^t) + \min_{\theta \in \Theta} \frac{1}{T}\sum_{t=1}^{T}\tilde{\mathcal{L}}_2(\theta;\lambda^t) + \epsilon_\theta && \text{(from (45))} \\
&= \min_{\xi \in [0,1]^K, \theta \in \Theta} \frac{1}{T}\sum_{t=1}^{T}\tilde{\mathcal{L}}(\xi,\theta;\lambda^t) + \epsilon_\theta \\
&= \min_{\xi \in [0,1]^K, \theta \in \Theta} \tilde{\mathcal{L}}(\xi,\theta;\bar{\lambda}) + \epsilon_\theta && \text{(by linearity of $\tilde{\mathcal{L}}$ in $\lambda^t$)} \\
&\leq \max_{\lambda \in \mathbb{R}_+^K} \min_{\xi \in [0,1]^K, \theta \in \Theta} \tilde{\mathcal{L}}(\xi,\theta;\lambda) + \epsilon_\theta \\
&\leq g(\tilde{\theta}^*) + \epsilon_\theta. \tag{47}
\end{aligned}
$$

where the last step follows from Lemma 7.

We also have from (46):

$$\frac{1}{T}\sum_{t=1}^{T}\mathcal{L}(\xi^t,\theta^t;\lambda^t) \geq \frac{1}{T}\sum_{t=1}^{T}\mathcal{L}(\xi^t,\theta^t;\lambda') - \epsilon_\lambda. \tag{48}$$

Combining (47) and (48), we have for any $\lambda' \in \Lambda$:

$$\frac{1}{T} \sum_{t=1}^{T} \mathcal{L}(\xi^t, \theta^t; \lambda') \leq g(\tilde{\theta}^*) + \epsilon_\theta + \epsilon_\lambda. \tag{49}$$

Setting $\lambda' = 0$ in (49) gives us:

$$\mathbf{E}_{\theta \sim \bar{\mu}}[g(\theta)] \leq g(\tilde{\theta}^*) + \epsilon_\theta + \epsilon_\lambda.$$

This completes the proof of optimality.

**Feasibility.** Recall that there are two sets of constraints $\phi^j(R_k(\mu)) \leq 0, \forall j \in [J]$ and $R_k(\mu) \leq \xi_k, k \in [K]$. We first look at the first set of constraints. Let $j' \in \operatorname{argmax}_{j \in [J]} \phi^j(\bar{\xi})$ and set $\lambda'_{j'} = \kappa$ and $\lambda'_j = 0, \forall j \neq j', j \in [J + K]$ in (49) (note that $\lambda' \in \Lambda$). This gives us:

$$\mathbf{E}_{\theta \sim \bar{\mu}}[g(\theta)] + \frac{\kappa}{T} \sum_{t=1}^{T} \phi^{j'}(\xi^t) \leq g(\theta^*) + \epsilon_\theta + \epsilon_\lambda.$$

By convexity of $\phi^j$ and using $g(\tilde{\theta}^*) \leq B_g$, we have:

$$\phi^{j'}(\bar{\xi}) \leq (B_g - \mathbf{E}_{\theta \sim \bar{\mu}}[g(\theta)] + \epsilon_\theta + \epsilon_\lambda)/\kappa \leq (B_g + \epsilon_\theta + \epsilon_\lambda)/\kappa$$

which implies:

$$\max_{j \in [J]} \phi^j(\bar{\xi}) \leq (B_g + \epsilon_\theta + \epsilon_\lambda)/\kappa.$$

Applying Lemma 5 to each $\phi^j$, we further get

$$\max_{j \in [J]} \phi^j(\mathbf{R}(\bar{\mu})) \leq L \max_{k \in [K]} (R_k(\bar{\mu}) - \bar{\xi}_k)_+ + (B_g + \epsilon_\theta + \epsilon_\lambda)/\kappa. \tag{50}$$

For the second set of constraints, let $k' \in \operatorname{argmax}_{k \in [K]} (R_k(\bar{\mu}) - \bar{\xi}_k)$. If $R_{k'}(\bar{\mu}) - \bar{\xi}_{k'} \leq 0$, then $\max_{k \in [K]} (R_k(\bar{\mu}) - \bar{\xi}_k)_+ = 0$. Otherwise, set $\lambda'_{J+k'} = \kappa$ and $\lambda'_j = 0, \forall j \neq J + k'$ in (49), and following the same steps as above, we get:

$$R_{k'}(\bar{\mu}) - \bar{\xi}_{k'} \leq (B_g + \epsilon_\theta + \epsilon_\lambda)/\kappa,$$

which further gives us:

$$\max_{k \in [K]} (R_k(\bar{\mu}) - \bar{\xi}_k)_+ = \max_{k \in [K]} (R_k(\bar{\mu}) - \bar{\xi}_k) \leq (B_g + \epsilon_\theta + \epsilon_\lambda)/\kappa. \tag{51}$$

Substituting (51) back in (50), we have:

$$\max_{j \in [J]} \phi^j(\mathbf{R}(\bar{\mu})) \leq (L + 1)(B_g + \epsilon_\theta + \epsilon_\lambda)/\kappa,$$

which completes the feasibility proof. $\qquad\square$

## K.2 Corollary for OGD on $\theta$ and $\lambda$

*Proof of Theorem 4.* We show that the iterates of Algorithm 2 form an approximate coarse-correlated equilibrium and apply Theorem 12.

The best-response strategy of the $\xi$-player gives us:

$$\frac{1}{T} \sum_{t=1}^{T} \mathcal{L}_1(\xi^t; \lambda^t) = \frac{1}{T} \sum_{t=1}^{T} \min_{\xi \in [0,1]^K} \mathcal{L}_1(\xi; \lambda^t) \leq \min_{\xi \in [0,1]^K} \frac{1}{T} \sum_{t=1}^{T} \mathcal{L}_1(\xi; \lambda^t).$$

We then derive no-regret guarantees for the updates of the $\theta$- and $\lambda$-player using standard convergence analysis for OGD [41] and standard arguments to convert an uniform online guarantee to a stochastic one [50]. For the sequence of losses $-\mathcal{L}(\xi^1, \cdot; \lambda^1), \ldots, -\mathcal{L}(\xi^T, \cdot; \lambda^T)$ optimized by the $\theta$-player, setting $\eta = \frac{B_\Theta}{B_\theta \sqrt{2T}}$ in Algorithm 2, we get the following regret bound for the $\theta$-player (see Corollary

3 in [23] for the complete derivation). With probability $\geq 1 - \delta/2$ over draws of stochastic gradients of $\tilde{\mathcal{L}}_1$:

$$\frac{1}{T}\sum_{t=1}^{T}\tilde{\mathcal{L}}(\xi^t, \theta^t; \lambda^t) \;\leq\; \min_{\theta \in \Theta}\frac{1}{T}\sum_{t=1}^{T}\tilde{\mathcal{L}}(\xi^t, \theta; \lambda^t) \;+\; 2B_\Theta\, B_\theta\sqrt{\frac{1 + 16\log(2/\delta)}{T}},$$

Similarly, for the sequence of losses $-\mathcal{L}(\xi^1, \theta^1; \cdot), \dots, -\mathcal{L}(\xi^T, \theta^T; \cdot)$ optimized by the $\lambda$-player, setting $\eta = \frac{\kappa}{B_\lambda\sqrt{2T}}$ in Algorithm 2, we get the following regret bound. With probability $\geq 1 - \delta/2$ over draws of stochastic gradients of $\mathcal{L}$:

$$\frac{1}{T}\sum_{t=1}^{T}\mathcal{L}(\xi^t, \theta^t; \lambda^t) \;\geq\; \max_{\lambda \in \Lambda}\frac{1}{T}\sum_{t=1}^{T}\mathcal{L}(\xi^t, \theta^t; \lambda) \;-\; 2\kappa\, B_\lambda\sqrt{\frac{1 + 16\log(2/\delta)}{T}}.$$

An application of Theorem 12 with $\kappa = (L+1)T^\omega$, for $\omega \in (0, 0.5)$ then gives us w.p. $\geq 1 - \delta$ over draws of stochastic gradients of $\tilde{\mathcal{L}}_1$ and $\mathcal{L}$:

$$\mathbf{E}_{\theta \sim \bar\mu}\left[g(\theta)\right] \;\leq\; g(\tilde\theta^*) \;+\; 2B_\Theta\, B_\theta\sqrt{\frac{1 + 16\log(2/\delta)}{T}} \;+\; 2(L+1)B_\lambda\frac{\sqrt{1 + 16\log(2/\delta)}}{T^{1/2-\omega}}$$

and

$$\max_{j \in [J]}\phi^j(\mathbf{R}(\bar\mu)) \;\leq\; \frac{B_g}{T^\omega} \;+\; 2B_\Theta\, B_\theta\frac{\sqrt{1 + 16\log(2/\delta)}}{T^{1/2+\omega}} \;+\; 2(L+1)B_\lambda\sqrt{\frac{1 + 16\log(2/\delta)}{T}},$$

which completes the proof. $\qquad\square$

## K.3 Discussion on Convergence Rate

Unlike Theorem 3, where we were able to achieve a $\mathcal{O}(1/\sqrt{T})$ convergence rate for both the objective and constraint violations, here we get a poorer rate of $\mathcal{O}(1/T^{1/2-\omega})$ for the objective and $\mathcal{O}(1/T^\omega)$ for the constraints. The choice of $\omega \in (0, 0.5)$ strikes a trade-off between the tightness of the two bounds. The reason we were able to get a better rate with Algorithm 1 is that we could use the assumption that the constraints can be satisfied with a margin $\gamma$ to set the radius of Lagrange multipliers $\kappa$ to a constant. This does not work with Algorithm 2 because of the mismatch in objectives optimized by the $\theta$- and $\lambda$-players, and hence we set $\kappa$ to a function that decreases with $T$.

## Footnotes

[3]https://github.com/google-research/tensorflow_constrained_optimization/