[Reviews · NeurIPS 2019]

Reviewer 1



Originality: -Compared to Cotter et al, they have shown more classes of problems (i.e. rate metric objective & constraint, sum of ratios) can be reduced to finding a saddle point. -In terms of theoretical techniques of solving the problem, I am not exactly sure (due to my confusion which I describe more further below) exactly what is novel and what is not, especially compared to Cotter et al. My understanding is that even with surrogate approaches, the authors of this paper have shown that there's a way to get convergence by having both max and min players play no regret algorithms (with respect to true lagragian and surrogate lagrangian respectively). In Cotter et al, the non-zero sum issue can be side-stepped with one player playing to minimize no-swap regret. Quality: -Overall, the paper flows nicely with sections divided up to account for different problems (rate metric, objective & constraint, sum of ratios). Also, most of proofs in the appendix seem to to be good. -Also, the empirical evaluations of the algorithm for different settings and on different datasets shows practicality of the approach and should be given credit. Clarity: -Personally, 'three player' game seems a little misleading in that it's rather that for the min player (i.e. the player who's trying to minimize the Lagrangian), the objective function is separable, so it can be minimized by minimizing L_1 and L_2 separately. Essentially, these two players trying to minimize the Lagrangian should be viewed simply as one player. Even when the surrogate functions are used, the minimizing player is trying to minimize L_1 + tilde(L)_2. -I don't quite understand the connection to Cotter et al. In both of these papers, the problem is when the constraints are non-convex and surrogate functions need to be used. In my understanding, Cotter et al's approach seems to be that it provides a proxy Lagrangian (instead of the real constraints, it uses the surrogate) where one player is playing no-regret while the other's trying to minimize no swap regret; the primal player is trying to minimize a proxy Lagrangian that involves the surrogate constraints, whereas the dual player is trying to maximize Lagragian with respect to the true constraints. In this paper, it's shown that one can do a similar thing through tilde(L)_2 (instead of real constraints, it uses the surrogate), but simply having both players play no-regret converges to approximately optimal solution. In this sense, my understanding is that proxy Lagrangian and tilde(L)_2 both serve the same purpose and the only difference is how they get the approximately optimal solution. For this reason, with my understanding (which may be wrong), I think it's more fitting to say that a different approach has been proposed than to say that an 'open problem' has been solved. Significance: -I think their results that that dynamics between a primal and dual player can solve many more constrained optimization problem involving non-convex functions are significant. Also, their unifying approach to show how previous works (SPADE, NEMSIS, etc) can be re-derived and/or be improved upon (appendix B, C) is quite nice, as in the future, one can turn to the given framework to solve the problems. Even just the fact that these problems can be reduced to this approached should be credited. However, this significance of the framework should also be qualified by noting that there are some previous papers that have already provided a similar framework. -Showing that both players playing no-regret with respect to true constraint and surrogate constraint respectively to attain an approximately optimal solution is of theoretical importance. -Availability of the tensor flow code will be of significance, too. Typos: -Right after line 485, right parenthesis is missing. -Line 568. Surrogate functions are not used in the oracle-based approach. -Line 578 "first state a lemma that* relate[s]" ********** POST REBUTTAL ************ They have clarified the paper's relationship to Cotter et al and said that they will add some clarifications in the main body and the appendix. Hence, my confusion about this has been addressed pretty well, so I still vote for accept.

Reviewer 2



The paper is well-written. See answers to question 1 for comments on originality etc.

Reviewer 3



Overall the paper provides new and insightful approach to tackle learning problems with non decomposable performance metrics. The result generalizes some existing approaches to optimizing non decomposable metrics and give provable algorithms for more general objective functions or constraints made of classification rates. The authors also resolve open questions from earlier work of Cotter et al (ALT 2019, JMLR 2019) about using OGD on true and surrogate Lagrangian to achieve convergence as a positive result although receiving a slightly worse convergence rate.) The problem of optimizing non decomposable metrics is significant and this paper provides algorithms with theoretically guarantees for a class of such learning problems and thus the results are themselves significant.

[Author Response · NeurIPS 2019]

We thank the reviewers for the positive comments and useful feedback. We provide responses to the main comments.

**Connections to Cotter et al:** There are two main differences between our paper and Cotter et al. (2019a;b):

1. Cotter et al. consider constrained optimization problems where the objective and constraints are linear functions of rates, i.e. are expected losses over a subset of the instance space. We handle a much broader class of problems where the objective and constraints can be general *non-decomposable* functions of rates including e.g. F-measure, KLD, etc.

2. Like us, Cotter et al. propose a formulation where the $\theta$-player optimizes the true Lagrangian and the $\lambda$-player optimizes a surrogate-approximation to the Lagrangian. They consider two algorithms, one where both the $\theta$- and $\lambda$-player seek to minimize (external) regret, and the other where the $\theta$-player optimizes alone minimizes external regret, while the $\lambda$-player minimizes swap regret. They are however able to show convergence guarantees only for the second algorithm, which arguably has a more complicated set of updates (requiring computing eigenvectors). They had acknowledged this saying they had "no theoretical justification" for their external regret algorithm (p. 27 in [2]).

We address that *lack of theoretical justification* (and will change to that wording rather than referring to it as an *open problem*, in accordance with Reviewer 1's comment) by providing a theoretically justified external regret algorithm in that we show that having both the $\theta$- and $\lambda$-players optimize external regret does indeed lead to convergence, and show this result for a more general setting than the one in Cotter et al. We will clarify this better in the main text and elaborate on this in Appendix D.

**Two vs Three-Player Viewpoint:** We agree that our formulation can be also be phrased as a two player game where the min-player plays uses different strategies for the $\xi$- and $\theta$- portions of the objective. While we find the three-player viewpoint to be a useful way to think about the problem algorithmically in that the three sets of parameters can use different optimization algorithms (see Table 1), this viewpoint is not crucial to the main contribution of the paper. We will clarify early on in the main text that our formulation can equivalently be regarded as a two-player game, and that the three-player viewpoint is an algorithmic perspective.

**Code:** We will make Tensorflow code available.

**Motivation for Surrogates.** Generalized rate metrics such as the G-mean or F-measure are non-convex and non-continuous functions of the model parameters, and hence directly optimizing them is hard in general. The traditional approach to optimizing non-continuous metrics is to work with convex surrogates that upper bound the metrics. For standard metrics such as the error rate, this is straight-forward and amounts to replacing the indicator function in the metrics with e.g. the hinge loss. This simple approach can however be problematic for more general non-decomposable metrics of the form $\psi\big(R_1(\theta), \ldots, R_K(\theta)\big)$, where $\psi$ is often defined in a restricted domain (e.g. square-root or KL-divergence), and replacing the rate terms with convex/concave relaxations may render $\psi$ undefined. We explain this with an example in Appendix C.

The proposed framework provides a cleaner solution for optimizing generalized rate metrics by using surrogates only where necessary. In particular, we formulate a max-min Lagrangian optimization problem, and use surrogate approximations for the Lagrangian when optimizing over the model parameters, and use the original rate values $R_k$'s for all other updates. We will include a discussion on surrogates in Section 2.

# References

[1] Cotter, A., Heinrich J., and Sridharan, K. (2019). Two-Player Games for Efficient Non-Convex Constrained Optimization. *Algorithmic Learning Theory*.

[2] Cotter, A., Jiang, H., Wang, S., Narayan, T., Gupta, M., You, S., and Sridharan, K. (2019). Optimization with Non-Differentiable Constraints with Applications to Fairness, Recall, Churn, and Other Goals. ArXiv:1809.04198. URL: https://arxiv.org/abs/1809.04198.


[Meta-Review · NeurIPS 2019]

The paper was very well received. Key strengths of the paper were its analysis of the three player game involving model parameters, Lagrange variables and slack variables and how it answered open problems from earlier work in the area. The core idea of "solve learning problems with functions of classification rates using a game" was appreciated. The response addressed some lingering questions about the relation to prior work (another success of the author response process!)